# Non-uniform temporal scaling of developmental processes in the mammalian cortex

Annalisa Paolino[1,2,6], Elizabeth H. Haines[1,2,6], Evan J. Bailey [1,2], Dylan A. Black[2], Ching Moey[2], Fernando García-Moreno [3,4], Linda J. Richards [1,2,5], Rodrigo Suárez [1,2] ✉ & Laura R. Fenlon [1,2] ✉

The time that it takes the brain to develop is highly variable across animals. Although staging systems equate major developmental milestones between mammalian species, it remains unclear how distinct processes of cortical development scale within these timeframes. Here, we compare the timing of cortical development in two mammals of similar size but different developmental pace: eutherian mice and marsupial fat-tailed dunnarts. Our results reveal that the temporal relationship between cell birth and laminar specification aligns to equivalent stages between these species, but that migration and axon extension do not scale uniformly according to the developmental stages, and are relatively more advanced in dunnarts. We identify a lack of basal intermediate progenitor cells in dunnarts that likely contributes in part to this timing difference. These findings demonstrate temporal limitations and differential plasticity of cortical developmental processes between similarly sized Therians and provide insight into subtle temporal changes that may have contributed to the early diversification of the mammalian brain.

There is remarkable variation in the rates of development between mammals, ranging from the 11-day gestation of the Virginia opossum to 655 days in the Indian elephant[1,2]. Smaller mammals tend to grow more rapidly and reach developmental milestones faster than larger mammals[3], including the time required to develop a mature brain[4,5]. However, subtle differences in the timing of developmental sequences (i.e. heterochronies) can also occur and potentially lead to evolutionary change[6–8]. An intriguing example of this is the neocortex, which expanded extraordinarily in hominids compared to all other mammals, an outcome that is thought to be linked to changes in the developmental timing of brain and body growth[5,9,10].

Although the broadly conserved scaling of developmental timing with body and brain size is consistent within the two major groups of Therian mammals, eutherians (placentals) and marsupials, it is not equivalent between them. Marsupials tend to take around two to three times longer to reach developmental milestones of body and brain growth than equivalently sized eutherians[4,5,11,12], such that two species with approximately equivalent developmental time courses, e.g. the marsupial short-tailed opossum and eutherian cat (approximately 15 versus 17 days to reach 40% of maximum brain weight) have vastly different adult brain weights (8 g versus 28.4 g)[4]. This distinction is in accordance with estimates of cortical cell cycle length that are also around three times slower in marsupials[13]. Marsupials and eutherians share many neocortical features, including a six-layered neuronal organization, with these layers developing in an inside-out manner[14,15] and subserving similar functions and connecting to similar targets[16,17]. Despite this broad conservation, key neocortical features also differ between the two lineages, including the route of interhemispheric connections, with marsupials displaying the ancestral condition of midline crossing of axons exclusively via the anterior commissure,

[1]The University of Queensland, School of Biomedical Sciences, Brisbane QLD 4072, Australia. [2]The University of Queensland, Queensland Brain Institute, Brisbane QLD 4072, Australia. [3]Achucarro Basque Center for Neuroscience, Scientific Park of the University of the Basque Country (UPV/EHU), 48940 Leioa, Spain. [4]IKERBASQUE Foundation, María Díaz de Haro 3, 48013 Bilbao, Spain. [5]Present address: Washington University in St Louis School of Medicine, Department of Neuroscience, St Louis, MO 63108, USA. [6]These authors contributed equally: Annalisa Paolino, Elizabeth H. Haines. ✉e-mail: r.suarez@uq.edu.au; l.fenlon@uq.edu.au

whereas in eutherians the corpus callosum evolved as a new tract[18–20]. There is also evidence of a lower cortical cell density in marsupials[21–23], and a relative dearth or even absence of basal intermediate progenitor cells[14,23,24], although there have been conflicting reports about this[13,25], perhaps reflecting differences between species and/or methods of investigation.

We recently discovered that the differential timing of expression of the transcription factor SATB2 determines whether neocortical axons take a marsupial-like commissural route (through the anterior commissure) or a more eutherian-like commissural route (through the corpus callosum)[16], highlighting the relevance of developmental timing differences for neocortical development and evolution. A subsequent study comparing the developing cortical transcriptome of mice and marsupial dunnarts revealed that this trend is reflective of a generalized advanced molecular maturity in dunnarts at equivalent stages of early corticogenesis[26]. However, despite the finding that subtle changes in developmental timing may underlie large phenotypic changes in mammalian brain evolution[16], the specific aspects of development that are similarly or differentially timed in mammals with distinct developmental periods, such as marsupials and eutherians, remain unclear. For instance, whether the developmental processes of corticogenesis (e.g. cell birth, cell migration, axon extension) are equivalently temporally scaled between species, or whether a subset of processes disproportionately impact developmental rate is not known. Understanding the rules and limitations of timing in neocortical development will increase our insight into the subtle changes in temporal dynamics that affect adult brain form and function, both within and across species.

To begin to address this issue, we explored how the temporal dynamics of cortical development scale with altered ontogenetic tempos in eutherian mice and similarly sized marsupial dunnarts. We employed a staging system that equates dunnart and mouse development based on shared anchor-points such as eye development[27] (1 day of development in mice is equivalent to around 3 days in dunnarts; Supplementary Fig. 1), as well as methods to selectively label progression of neuronal specification, migration, and circuit development comparably in both species[16,28]. If each of the specific developmental processes we examined were to scale uniformly according to this staging system, then we would expect each to take around three times as long in dunnart versus mouse in absolute terms, and be equivalent when compared by developmental stage. However, our findings revealed that distinct developmental processes scale non-uniformly between the two species, with cell birth/specification aligning to the shared staging system, whereas migration and axon extension both occur earlier in dunnarts than mice by stage. We also identified the absence of proliferative basal intermediate progenitor cells in dunnarts as a possible feature that contributes to these differences.

## Results
### Conserved timing of cortical cell birth in mice and dunnarts
To assess whether the rate at which cortical cell birth contributes to the distinct layers of the cortex, we performed ethynyl deoxyuridine (EdU) birthdating in mice and dunnarts at equivalent stages (based on our published staging system[27], summarized in Supplementary Fig. 1a), and collected the brains at a single equivalent stage after the end of cortical neurogenesis (stage (S) 27). We first quantified the spatial gradient of cortical cell birth during development and confirmed previous qualitative reports that cell birth in marsupials proceeds in a rostrolateral to caudomedial gradient[29], as has been similarly demonstrated in eutherians[30–33] (Fig. 1a–d and Supplementary Fig. 1b, c; N and statistical tests for all data detailed in Supplementary Table 3). We then examined the timing of cell birth relative to lamination of the cortex (Fig. 1e, f). To determine whether the timing of cell birth corresponds to the identity and position of the final cortical layer, we measured the median of the distribution of EdU⁺ cells within the primary

somatosensory cortex (S1) as a percentage of cortical width weighted to the equivalent size of cortical layers between species. This analysis revealed no significant differences in the pattern of cell birth and neuron specification/lamination between species (Fig. 1g). The proportion of upper layer (layer (L)2-4; UL) to deep layer (L5/6; DL) cells at each stage of injection also exhibited equivalent temporal dynamics of neurogenesis, with the switch between DL and UL cell birth occurring at S21 in both species (Fig. 1h). Interestingly, the ratios of UL/DL EdU⁺ cells were significantly more extreme in mice than dunnarts for the periods of both DL and UL cell birth, indicating that birthdate may be more specifically related to final laminar position in mice (Fig. 1h).

To further compare trends in cell birth, we calculated the total density of EdU⁺ cells across the cortex of each species and found that it was greater in mice that were injected with EdU at earlier stages corresponding to DL birth, and higher for dunnarts at later stages corresponding to UL birth (Fig. 1i). We then examined whether this distinction relates to the final adult cortical organization by quantifying the size and relative proportion of cortical layers, as well as cell densities, in the adult cortex of both species. Dunnarts have a smaller cortical width (from ventricle to pia), consistent with their slightly smaller adult brain and body size compared to mice[34] (Fig. 1j–l). However, consistent with the difference in developmental cell density illustrated in Fig. 1i, the DLs occupied a greater proportion of the total cortical width in the adult mouse cortex, whereas the ULs occupied a greater proportion in dunnarts (Fig. 1m), similar to previous reports in opossums and wallabies[23]. We also found an overall difference in the cell density of the adult cortex between species, likely driven by a significantly lower density in the intermediate zone and L6 in dunnarts (although non-significant lower trends in dunnart cell densities occurred throughout cortical layers; Fig. 1n).

### Inter-species cell maturation heterochrony relative to stage
Given the correspondence between stages of cell birth and laminar position between species (Fig. 1), we next explored whether the timing of neuronal maturation is also conserved using the same staging system. We electroporated mice (in utero) and dunnarts (in pouch) with a fluorophore at either S20 or S23, to label DL or UL neurons, respectively[28], and collected the brains across a range of subsequent stages (Fig. 2a–d and Supplementary Fig. 2). Quantification of the leading edge of the electroporated cell population revealed that dunnart cells were significantly more advanced through the cortical plate (CP) than mouse cells for the first two stages post-transfection for S20 (Fig. 2b), and S23 (Fig. 2d) electroporations, after which there was no difference. This trend was also reflected in the mean position of the 2% most advanced electroporated cells in the cortex (Supplementary Fig. 2c, f). We also quantified the proportion of electroporated cells occupying each cortical layer and found that, for both S20 and S23 electroporations, dunnart neurons born at an equivalent stage to mouse neurons were located in the more mature compartments of the brain in significantly higher proportions during the initial stages, with the two species reaching equivalent proportions four to five stages later (Fig. 2e, f).

To determine whether this advanced location truly correlates with a more mature cell state in dunnarts, we performed electroporations at an intermediate stage (S21), collected tissue one stage later, and analyzed the degree of colocalization of electroporated TdTom⁺ cells with expression of the immature marker SOX2 and the predominantly postmitotic neuronal marker NEUROD1. We found that a significantly higher ratio of electroporated cells colocalized with NEUROD1 than with SOX2 expression in dunnarts compared with mice (Fig. 2g–i), confirming an advanced maturation state relative to stage in dunnarts.

### Neuronal maturation by day is slower in dunnarts than mice
Our results indicate a faster migration and maturation of electroporated cortical cells in dunnarts than mice relative to developmental

 

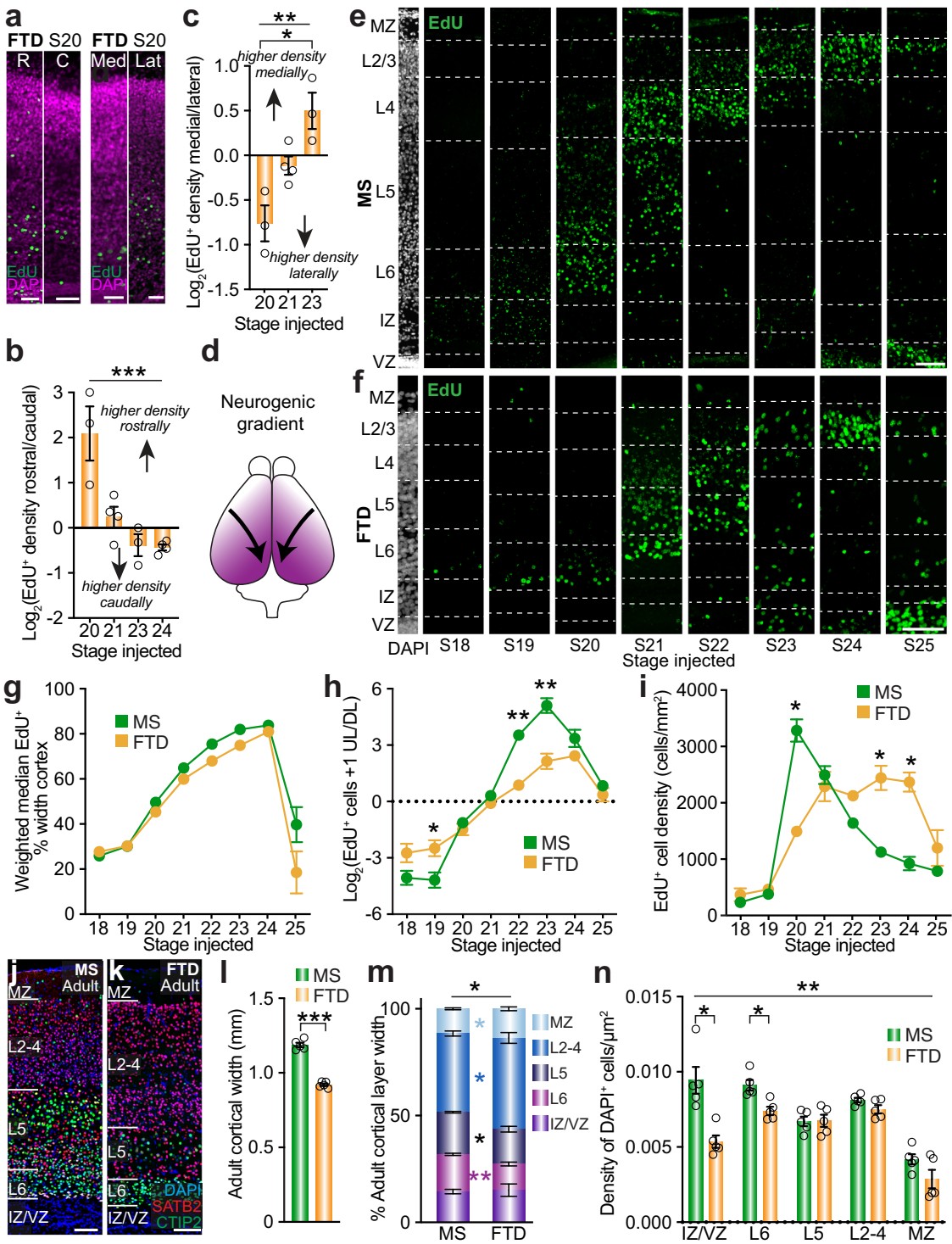

stage. However, in absolute time (days) dunnarts take longer to develop than mice (2–3 days per stage in dunnart versus 1 day in mice; Supplementary Fig. 1a). We therefore next compared neuronal migration between species relative to absolute days (not developmental stages) to gain insight into its dynamics. An analysis of the leading edge of the labeled cells electroporated at S20 and S23 (labeling DLs and ULs, respectively) and collected on subsequent days revealed that dunnart cells were less advanced than mouse cells until day three after S20 electroporation (Fig. 3a), and until day five after S23 electroporation (Fig. 3b and Supplementary Fig. 3a–d). We next colabeled sections with classic markers of cortical development cell types: PAX6 (apical progenitor cells), TBR2 (subventricular zone, SVZ cells)

and TBR1 (newly differentiated neurons) and plotted the distribution of TdTom+ cells across the cortical width on the days following electroporation, quantifying the percentage of total cells located in each of the compartments (Fig. 3c–j and Supplementary Fig. 3e–p). In accordance with the leading-edge quantification (Fig. 3a, b), this analysis revealed more advanced positioning of cells in mice than dunnarts per day, with dunnarts taking around two days to reach similar proportions to one day of mouse development (Fig. 3d and Supplementary Fig. 3o).

In order to understand the comparative progression of newborn cells, we examined their morphology in the days following transfection. In eutherians, the morphology of cortical cells follows a

**Fig. 1 | Mice and dunnarts share a conserved spatiotemporal progression of neocortical cell birth relative to developmental stage. a** Rostral versus caudal (left) and medial versus lateral (right) areas of the dunnart neocortex following EdU injection at S20 and collection at S27. **b, c** EdU-labeled cell density in rostral versus caudal (**b**) or medial versus lateral (**c**) areas of the dunnart cortex following EdU injection between S20-24 and collection at S27. N of S20:3, S21:4, S23:3, S24:4. **d** Schematic illustrating the directionality of the dunnart cortical neurogenic gradient. **e, f** Primary somatosensory cortex depicting EdU⁺ cell distribution across the cortex following injection between S18-25 and collection at S27 in mice (**e**) and dunnarts (**f**). **g** Weighted median of EdU⁺ cell distribution as a percentage of cortical width at each stage of injection. N for MS S18-S25:6; FTD S18:4, S19:5, S20:4, S21:6, S22:3, S23:5, S24:4, S25:3. **h** Proportion of EdU⁺ cells in the UL versus DL represented as Log₂ across injected stages. N = (**g**). **i** Quantification of mean EdU⁺ cell densities

throughout the cortical width in mice and dunnarts across injected stages. N = (**g**). **j, k** Adult mouse (**j**) and dunnart (**k**) primary somatosensory cortex stained with DAPI, SATB2, CTIP2. **l** Adult cortical width in each species. *N* for both:5. **m** Adult layers width as a proportion of the entire cortical width in mice and dunnarts. N = (**l**). **n** Quantification of DAPI⁺ adult cells density within each cortical layer. N = (**l**). Data are presented as mean values ±SEM, and were compared using one-way ANOVAs followed by pairwise *t*-tests (**b, c**), or Mann–Whitney *U* tests (**g–l**), MANOVA followed by pairwise log ratio comparisons (**m**) or Aligned Rank Test followed by pairwise Mann-Whitney *U* tests (**n**); *$p < 0.05$, **$p < 0.01$, ***$p < 0.001$. See Supplementary Table 3 for exact *p* values and statistical test details. C caudal, DL deep layers, FTD fat-tailed dunnart, IZ intermediate zone, Lat lateral, L layer, Med medial, MS mouse, MZ marginal zone, R rostral, S stage, UL upper layers, VZ ventricular zone. Scale bars: **a** = 50 µm; **e, f, j, k** = 100 µm.

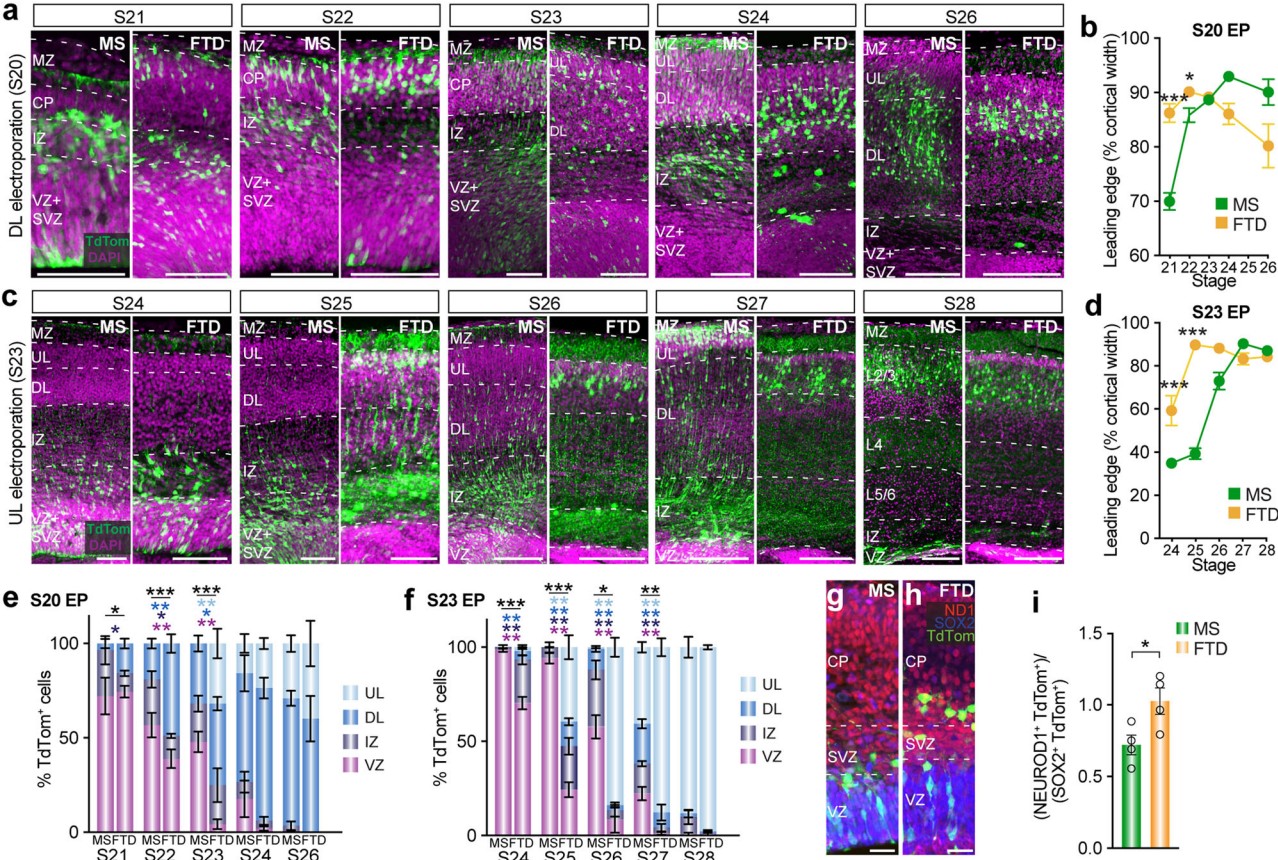

**Fig. 2 | Neuronal maturation and migration are relatively more advanced in dunnarts than mice. a, b** Cortical images (**a**) and quantification of leading edge of migrating cells distribution (**b**) in the primary somatosensory cortex of mice and dunnarts electroporated at S20 (labeling DL neurons) and collected between S21 and S26. N for MS S21:5, S22:5, S23:9, S24:3, S26:3; FTD S21:20, S22:13, S23:3, S24:3, S26:3. **c, d** Cortical images (**c**) and quantification of the leading edge of migrating cells distribution (**d**) in the primary somatosensory cortex of mice and dunnarts electroporated at S23 (labeling UL neurons) and collected between S24 and S28. N for MS S24:9, S25:8, S26:5, S27:4, S28:4; FTD S24:7, S25:9, S26:4, S27:5, S28:7. **e, f** Quantification of the proportion of electroporated TdTom⁺ cells in each compartment of the cortex following electroporation at S20 (**e**) or S23 (**f**) and subsequent collection across a range of developmental stages. N for MS in (**e**) S21:5, S22:5, S23:9, S24:3, S26:3; FTD in (**e**) S21:21, S22:13, S23:3, S24:3, S26:3. N for MS in

(**f**) S24:9, S25:8, S26:5, S27:4, S28:4; FTD in (**f**) S24:7, S25:9, S26:4, S27:5, S28:7. **g, h** Images of mouse (**g**) and dunnart (**h**) cortex electroporated at S21 and collected at S22 colabeled with NEUROD1 (ND1, red) and SOX2 (blue). **i** Quantification of colabeled TdTom⁺NEUROD1⁺ cells as a proportion of the colabeled TdTom⁺SOX2⁺ cells in the cortex of each species electroporated at S21 and collected at S22. N for MS and FTD:4. Data are presented as mean values ± SEM and were compared using Mann-Whitney *U* tests (**b, d**), MANOVA or Npmv followed by pairwise log ratio comparisons (**e, f**) or *t*-test (**i**); *$p < 0.05$, **$p < 0.01$, ***$p < 0.001$. See Supplementary Table 3 for exact *p* values and statistical test details. CP cortical plate, DL deep layers, FTD fat-tailed dunnart, IZ intermediate zone, L layer, MS mouse, MZ marginal zone, S stage, SVZ subventricular zone, UL upper layers, VZ ventricular zone. Scale bars: **a, b** S21-25 = 100 µm; S26-28 = 200 µm; **g, h** = 25 µm.

progression from bipolar morphology as radial glia in the ventricular zone (VZ), to multipolar morphology as they proliferate or migrate through the SVZ, after which they revert to a bipolar morphology as they resume migration into the cortical plate[35,36]. The morphology of electroporated cells in mice and dunnarts showed similar trends to

those illustrated in Fig. 3a–d, with mouse cells already being located in the TBR2-positive SVZ by 1 day post-electroporation, displaying characteristic multipolar morphology (Fig. 3e, e′), whereas dunnart cells were still bipolar and located in the VZ one day after electroporation (Fig. 3g, g′). However, day-two dunnarts resembled day-one mice, with

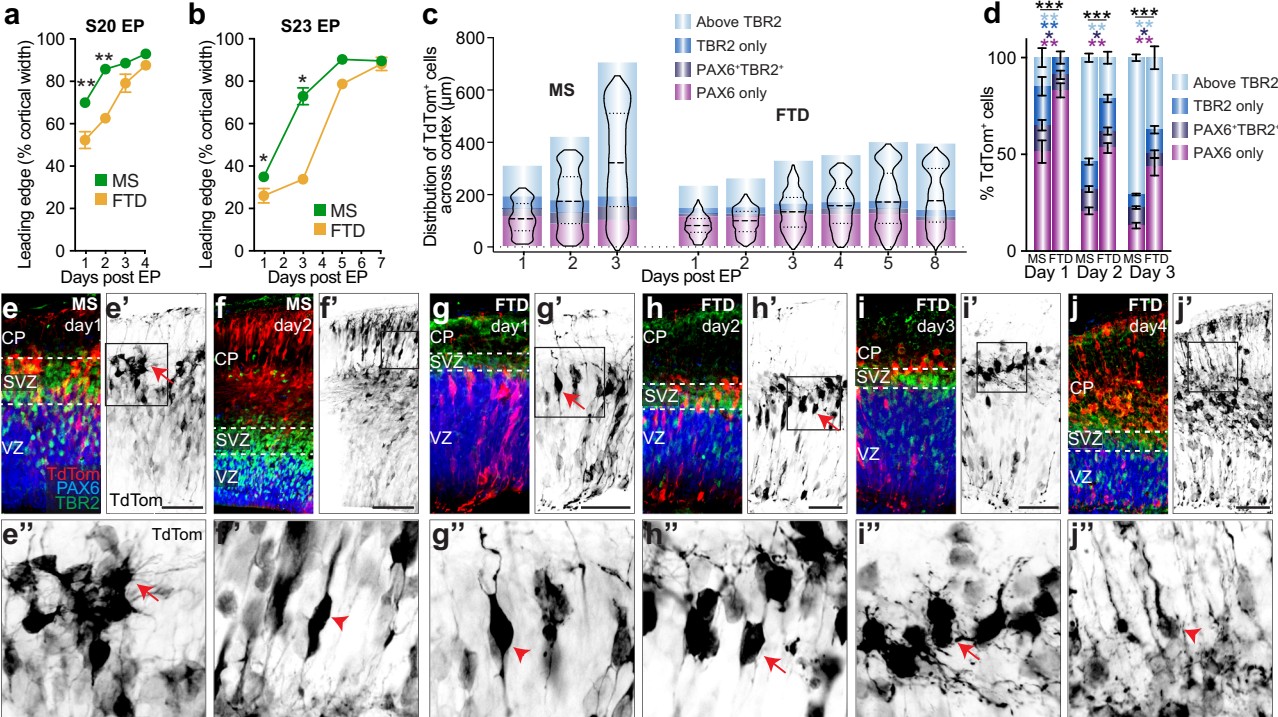

**Fig. 3 | Neuronal migration and maturation is faster in mice than dunnarts relative to absolute days. a, b** Quantification of the leading edge of the TdTom-labeled cell population as it migrates through the cortex following electroporation at S20 (**a**) or S23 (**b**). N for MS in (**a**) Day1:5, Day2:5, Day3:9, Day4:3; FTD in (**a**) Day1:7, Day2:8, Day3:6, Day4:8. N for MS in (**b**) Day1:9, Day3:5, Day5:4, Day7:9; FTD in (**b**) Day1:3, Day3:4, Day5:3, Day7:3. **c** Violin plots showing the accumulated distribution of TdTom+ cells in mouse and dunnart cortices following electroporation at S20 and collection at a range of days subsequently. Immunohistochemistry against TBR2 and PAX6 demarcates the borders of molecularly defined compartments throughout the cortex (the upper limit of the "Above TBR2" compartment is the average border of the top of the cortex at each age of collection). N for MS Day1:5; Day2:7; Day3:16; FTD Day1:11; Day2:13; Day3:13; Day4:9; Day5:12; Day8:12. **d** Quantification of the proportion of TdTom+ cells in each immunolabeled band of

mice and dunnarts from 1–3 days post-electroporation at S20. N for MS Day1:5, Day2:7, Day3:16; FTD Day1:11, Day2:13, Day3:13. **e–j** Cortical images of TdTom+ cells following electroporation at S20 and collection on subsequent days in mice (**e–f**) and dunnarts (**g–j**), colabeled with antibodies against PAX6 and TBR2. Arrows demarcate multipolar cells whereas arrowheads demarcate bipolar cells. Data are presented as mean values ±SEM in all the graphs, except for **c**, which is presented as the median with the first and third quartiles, and were compared using Mann–Whitney U tests (**a, b**) or Npmv followed by pairwise log ratio comparisons (**d**); *$p < 0.05$, **$p < 0.01$, ***$p < 0.001$. See Supplementary Table 3 for exact p values and statistical test details. CP cortical plate, DL deep layers, EP electroporation, FTD fat-tailed dunnart, MS mouse, SVZ subventricular zone, UL upper layers, VZ ventricular zone. Scale bars: **e′–j′** = 50 μm; **e″–j″** = 25 μm.

multipolar cells located in the SVZ (Fig. 3h, h′). Similarly, day-four dunnarts resembled day-two mice, with bipolar neurons migrating and settling in the CP where they started to extend dendritic and axonal processes (Fig. 3f, f′, j, j′).

### Delayed progenitor cells are present in dunnarts

Given the advanced maturation of neurons relative to stage in dunnarts compared with mice, we next explored candidate causative mechanisms that might be driving these differences in timing. First, it has previously been hypothesized that a population of delayed progenitor cells identified in mice (labeled using an Emx2 enhancer), that ultimately contributes to callosal neurons, may be absent in marsupials, as it was found to be absent in chicken[37]. To test this hypothesis, we co-electroporated dunnarts at S20 with either a control (CAG-Cre) or Emx2-Cre construct, together with a nuclear localized Cre-responsive GFP plasmid and a CAG-TdTom plasmid to label all transfected cells (Fig. 4). We found no difference in the distribution of TdTom+ cells in either condition following collection one or eight stages later, indicating that the Emx2-Cre labeling construct does not alter the distribution of electroporated cells (Supplementary Fig. 4a, b). However, the localization of the GFP+ cells was significantly different between conditions, with more Emx2-Cre-labeled cells in the VZ than the CP, and more CAG-Cre cells in the CP than the VZ (Fig. 4a–c), indicating that Emx2-Cre specifically labels a delayed progenitor subpopulation in dunnarts, similar to previous findings in mice[37].

To examine where the progeny of these cells ultimately resides, we performed similar electroporations (with a cytoplasmic Cre-responsive GFP plasmid[38]), but collected the brains at a later stage (S28; P50; Fig. 4d–k). The distribution of GFP+ Cre-labeled cells again differed between conditions, with a larger proportion of Emx2-Cre cells residing in the ULs of the neocortex (Fig. 4d–f), similar to mice[37]. A significantly higher proportion of Emx2-Cre- than CAG-Cre-labeled cells in dunnarts was also SATB2+ (a marker of cortico-cortical projections in both mice and dunnarts;[16] Fig. 4g–i) and axons from this neuronal population could be visualized crossing the midline through the anterior commissure (Fig. 4j–k), in keeping with the identification of this population in eutherians as predominantly commissural[37]. These results suggest that dunnarts have a similar Emx2-Cre-labeled delayed progenitor population to that identified in mice, which ultimately predominantly populates the commissural ULs of the cortex, and that this is unlikely to be a mechanism that explains the timing differences between species.

### Dunnarts lack basal intermediate progenitor cells

We next investigated another candidate feature that may underlie the differential timing of maturation between species: the presence of basal intermediate progenitor cells as an intermediary step of neurogenesis[39]. This process is well-characterized across eutherians and is thought to underlie the remarkable expansion of the primate brain[40,41]. Based on observations of very few mitotic cells outside the

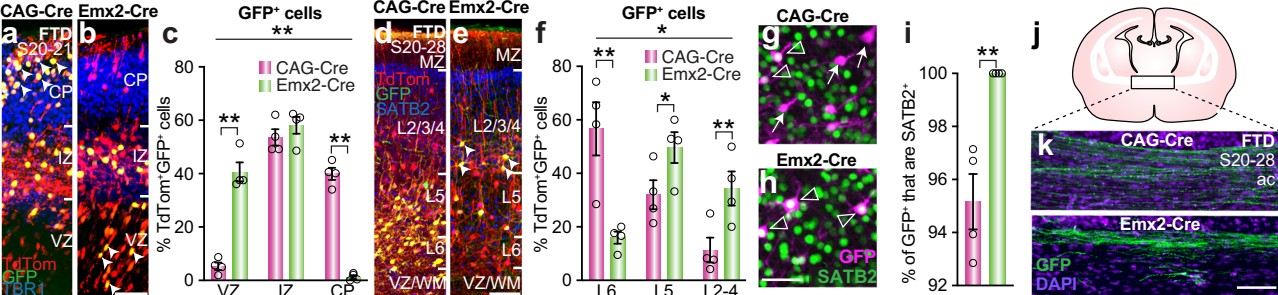

**Fig. 4 | Dunnarts have a delayed progenitor subpopulation that predominantly becomes upper layer commissural neurons. a, b** Cortical images of dunnart primary somatosensory cortex electroporated at P11 (S20) and collected at P16 (S21) with a combination of CAG-TdTom, PBCAG-STOP-H2BEGFP and either CAG-Cre (**a**) or Emx2-Cre (**b**) labeling. Arrowheads indicate examples of GFP+TdTom+ cells in areas that differ between conditions. **c** Quantification of the proportion of colabeled TdTom+GFP+ cells from P11-P16 electroporated brains in each compartment of the neocortex. N for both groups: 4. **d, e** Cortical images of dunnart primary somatosensory cortex electroporated at P11 (S20) and collected at P50 (S28) with a combination of CAG-TdTom, CAG-FloxedStop-GFP and either CAG-Cre (**d**) or Emx2-Cre (**e**) labeling. Arrowheads indicate examples of GFP+TdTom+ cells in areas that differ between conditions. **f** Quantification of the proportion of colabeled TdTom+GFP+ cells from P11-P50 electroporated brains in each layer of the neocortex. N = (**c**). **g, h** Magnified images of P11-P50 electroporations from d-e showing GFP-labeled electroporated cells with SATB2 immunolabeling and either coexpression (empty arrowheads) or no expression (arrows) in CAG-Cre (**g**) and Emx2-Cre (**h**) conditions. **i** Quantification of the proportion of GFP+ cells that are also SATB2+ following P11-P50 electroporation of the CAG-Cre or Emx2-Cre plasmid combinations from **d** to **h**. N = (**c**). **j, k** Schematics and representative images of the anterior commissure of dunnarts following P11-P50 electroporation of CAG-Cre (**j**) or Emx2-Cre (**k**) plasmid combinations. Data are presented as mean values ±SEM and were compared using MANOVA followed by log ratio comparisons (**c, f**) or t-test (**i**); *$p < 0.05$, **$p < 0.01$, ***$p < 0.001$. See Supplementary Table 3 for exact $p$ values and statistical test details. ac anterior commissure, CP cortical plate, D day, FTD fat-tailed dunnart, GFP green fluorescent protein, IZ intermediate zone, L layer, MZ marginal zone, S stage, VZ/WM ventricular zone/white matter. Scale bars: **a, b, d, e, g, h** = 50 µm; **j, h** = 100 µm.

VZ of developing opossums, it was initially suggested that marsupials lack a SVZ[24]. Later studies, using phospho-histone 3 (PH3) as a marker of mitotic cells in opossums, found a few cells above the VZ, albeit after the neurogenic period, further indicating that these might not correspond to true basal intermediate progenitor cells[14,23]. A more recent study in wallabies claimed that the presence of subventricular PH3+ cells was evidence of the existence of basal progenitors in marsupials. However, these cells represented less than 1% of all undifferentiated cells and peaked at stages when most of cortical neurogenesis had occurred[13]. Importantly, no studies to date that have described a putative cycling cell in the SVZ of a marsupial have definitively shown that it originated from the VZ. This is important, as other cell types such as microglia could be proliferating in this region and be misidentified as a basal progenitor cell.

To investigate whether cells from the VZ form cycling basal intermediate progenitor cells in dunnart, we electroporated both species at S20 with a fluorophore, collected brains on subsequent days of development, and colabeled sections with antibodies against the mitotic marker PH3 and the general cell-cycle marker PCNA (Fig. 5a–f). This protocol allows spatiotemporal quantification of proliferation specifically in cells that arise from the VZ. We then plotted the average distribution of the electroporated TdTom+ cells on an averaged cortical width divided into PCNA+ and PCNA- zones, as well as all PH3+ cells and TdTom+PH3+ cells for both species (Fig. 5g and Supplementary Fig. 5a). Whereas the PH3+ and TdTom+PH3+ distributions of mice clearly divided into two compartments on all three days post-electroporation, the equivalent dunnart populations remained confined to the VZ (Fig. 5g). Quantification of the density of all PH3+ mitotic cells in the VZ and SVZ (Fig. 5h), as well as the proportion of TdTom+ cells that were also PH3+ (Fig. 5i) in the VZ and SVZ of each species, revealed a significantly lower density of PH3+ mitotic cells, as well as a lower percentage of TdTom+PH3+ colabeled mitotic cells in the SVZ, but not the VZ across all three days post-electroporation. Notably, we did not record a single animal with a TdTom+PH3+ cell in the SVZ in dunnarts (Fig. 5i), leading us to question the identity of the rare PH3+ cells in the dunnart SVZ (Fig. 5h) that were never colabeled with TdTom (Fig. 5i). We found that there were significantly more PH3+ cells above the VZ in the electroporated hemisphere than the non-electroporated hemisphere of dunnarts for the first two days post-electroporation

(Supplementary Fig. 5b–g), and these were never TdTom+ (Fig. 5i). This led us to conclude that these were likely to be transiently recruited microglia, as has previously been reported subsequent to brain electroporations[42], underlining the importance of using techniques that trace lineages from the VZ such that other proliferating cell types are not misidentified as basal progenitor cells.

Given that we could not identify a basal intermediate progenitor cell population of VZ origin in dunnarts, we were curious about the identity of the TBR2+ band that we (Fig. 3) and others[14] have identified in marsupials. To address this, we immunolabeled S21 mouse and dunnart brains against TBR2 and PH3, and compared the proportion of TBR2+ cells that were also PH3+ (mitotic) between species. This analysis revealed that no TBR2+ cell was ever PH3+ in the dunnart SVZ, unlike in the mouse (Fig. 5j–l). This result could be due to differences in M phase length, leading to a lower likelihood of PH3+ labeling in one species compared to another. However, when we repeated this experiment with the general cell cycle marker PCNA, we found that the proportion of TBR2+ cells colabeled for PCNA was also significantly higher in mice than dunnarts (at or near 0 in all dunnarts; Fig. 5m–o). This result was not due to a relative paucity of PCNA labeling in dunnarts, as the proportion of SOX2+ cells (a marker of apical progenitors) that colabeled with PCNA was higher in the VZ of dunnarts than mice (Fig. 5p–r). Taken together, these findings indicate that dunnarts do not have a proliferative basal intermediate progenitor compartment, and that their TBR2+ population (which lies in the same position as the mouse SVZ) does not constitute basal intermediate progenitor cells. Instead, it is likely that this population comprises migrating post-mitotic neurons, as has previously been reported for TBR2+ cells in chicken and gecko[43–45].

## Conserved order of axon extension in mice and dunnarts

The differences in migration timing by equivalent stage that we observed between mice and dunnarts raises the question of whether concurrent/subsequent processes also display distinct dynamics between lineages. In eutherians, newly born pyramidal neurons begin to extend an axon collateral in the intermediate zone as they migrate towards the CP. In mice, the directionality of the initial axons changes as development proceeds, whereby axons first project laterally from the cortex (e.g. into the internal capsule), after which subsequent

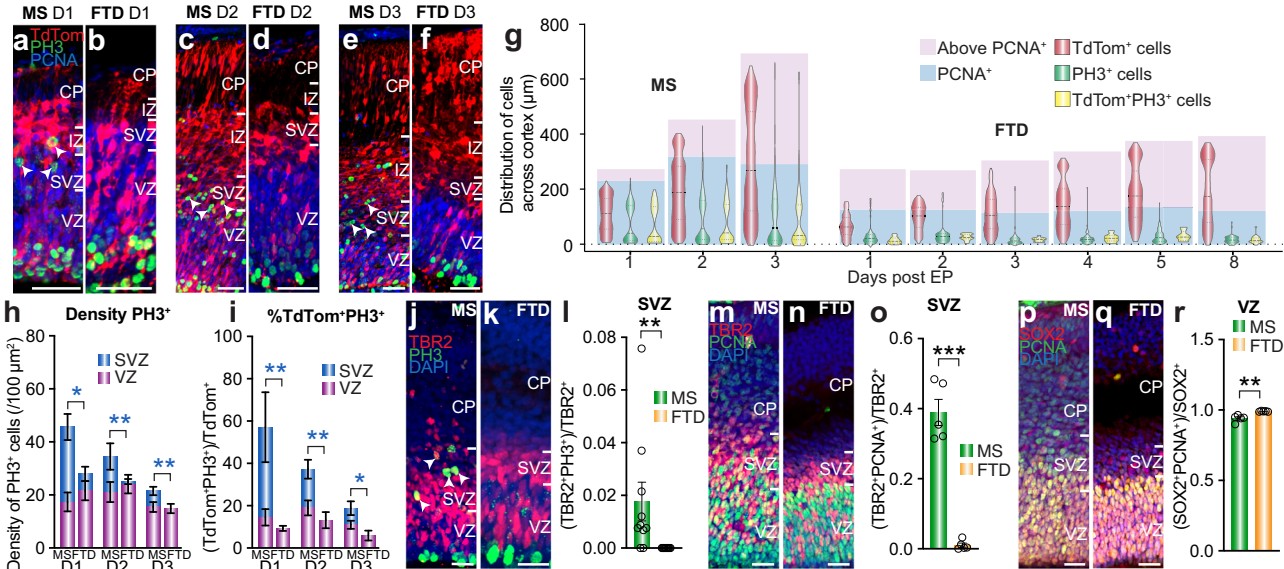

**Fig. 5 | Mitosing basal intermediate progenitor cells are absent in dunnarts.**
**a–f** Mouse (**a**, **c**, **e**) and dunnart (**b**, **d**, **f**) brains electroporated with CAG-TdTom at S20 and collected 1 (**a**, **b**), 2 (**c**, **d**), or 3 (**e**, **f**) days later and immunolabeled for PH3 and PCNA. **g** Violin plots showing aggregates of brains electroporated with TdTom and collected between 1 and 8 days later for mice (left) and dunnarts (right). Cell distributions across the cortical width are shown for TdTom[+], PH3[+], and TdTom[+]PH3[+] cells. The averaged boundaries of the PCNA[+] and PCNA[-] regions are shown underneath the plots. N for (**a–g**) MS Day1:6; Day2:7; Day3:13; FTD Day1:9; Day2:12; Day3:9; Day4:9; Day5:10; Day8:15. **h, i** Analysis of the brain series in (**a–f**) showing PH3[+] cells density (**h**) and proportion of TdTom[+] cells coexpressing PH3 (**i**) in the VZ and SVZ following electroporation at S20 and collection 1–3 days later. N for MS in (**h**) Day1:5; Day2:6; Day3:9; FTD in (**h, i**) Day1:7; Day2:7; Day3:6. For MS in (**i**) Day1:5; Day2:4; Day3:9. **j–l** S21 mouse (**j**) and dunnart (**k**) brains immunolabeled

for TBR2, PH3, and DAPI and quantification (**l**) of TBR2[+] cells coexpressing PH3 in the SVZ; arrowheads demarcate PH3[+] cells in the SVZ. N for MS:6; FTD:8. **m–o** S21 mouse (**m**) and dunnart (**n**) brains immunolabeled for TBR2, PCNA and DAPI and quantification (**o**) of TBR2[+] cells coexpressing PCNA in the SVZ. N for both:5. **p–r** S21 mouse (**p**) and dunnart (**q**) brains immunolabeled for SOX2, PCNA, and DAPI and quantification (**r**) of SOX2[+] cells coexpressing PCNA in the VZ. N for both:5. Data are presented as mean values ±SEM in all the graphs, except (**g**), which is presented as the median with the first and third quartiles, and were compared using *t*-tests or Mann-Whitney *U* tests where relevant; *$p < 0.05$, **$p < 0.01$, ***$p < 0.001$. See Supplementary Table 3 for exact *p* values and statistical test details. CP cortical plate, D day, EP electroporation, FTD fat-tailed dunnart, IZ intermediate zone, MS mouse, SVZ subventricular zone, VZ ventricular zone. Scale bars: **a–f** = 50 µm; **j, k, m, n, p, q** = 25 µm.

neurons project axons medially into the corpus callosum[46–48]. We have previously shown that, although all commissural axons project laterally through the anterior commissure in dunnarts, the adult brains have medial projections from S1 that project within the gray matter and terminate in the ipsilateral cingulate cortex[12]. A similar population of intracortical medially projecting neurons that are formed postnatally, after the establishment of the corpus callosum (between P2 and P8), has been demonstrated in mice[49,50]. This medial ipsilaterally projecting tract in dunnarts is an intriguing candidate that may have contributed physical and/or molecular cues that were adopted during the evolution of the corpus callosum in eutherians, and we therefore used it as an opportunity to compare the timing of medially versus laterally projecting axons between species, despite very different projection targets. To assess whether the medial and lateral projections of dunnarts follow the same temporal order as those of mice, we quantified the length of axons from an electroporated cell field in both species following DL electroporation. Our results revealed that, in both mice and dunnarts, lateral axons projected first, followed by medial axons, despite the differing targets of these projections between species (Fig. 6a–f). Further, the timing of this switch appears to be conserved for DL electroporated neurons, with the first clear medial projections emerging at S22 in both species (Fig. 6a–f).

A previous study in wallabies reported that axons from the piriform cortex project through the anterior commissure prior to neocortical commissural axons[51]. However, the axonal order of crossing for different areas within the neocortex has not been characterized for any marsupial. In mice, the order in which neocortical axons cross the corpus callosum follows a rostrolateral to caudomedial gradient[51,52], likely related to the same gradient of cell birth[30]. In dunnarts, in accordance with the conserved gradient of cell birth that we identified (Fig. 1a–d),

double electroporation experiments at S23 revealed that rostral neocortical neurons extend significantly longer axons than neurons from caudal regions by S24 (Fig. 6g, h), and that longer axons extend from more ventrolateral than dorsomedial electroporated cell fields (Fig. 6i). However, in mice, neocortical populations are not the first to form the corpus callosum, and this tract is instead pioneered by axons from the more dorsomedial cingulate cortex[53,54]. To examine whether this cingulate population also pioneers the neocortical portion of the anterior commissure in dunnarts, we injected DiI and DiD crystals into the cingulate cortex or neocortex of S21 (postnatal day (P)12) dunnart fixed brains (in order to anterogradely label developing axons) and analyzed the distance remaining for labeled axons from these populations to reach the midline. We found that a significantly greater distance remained for cingulate cortex axons than those from the neocortex (Fig. 6j–l), indicating that the cingulate cortex does not pioneer the dorsal pallial component of the anterior commissure in dunnarts, but rather their axons follow neocortical projections. This represents a major distinction in how these tracts develop between species (summary of findings in Fig. 6m, n). We also compared EdU birthdating for the cingulate cortex between mice and dunnarts and found that, compared to stage, the UL of the cingulate cortex (i.e. cortically projecting[53]) is born significantly later in dunnarts than in mice (Supplementary Fig. 6). This suggests an intriguing hypothesis whereby the accelerated development of cingulate neurons is related to the emergence of its novel role in pioneering the corpus callosum in eutherian mammals.

**Axon extension is more advanced in dunnarts relative to stage**
To compare the timing of axon extension between species, we electroporated the brains of mice or dunnarts at S20 and S23, collected the tissue at a range of subsequent developmental stages and quantified

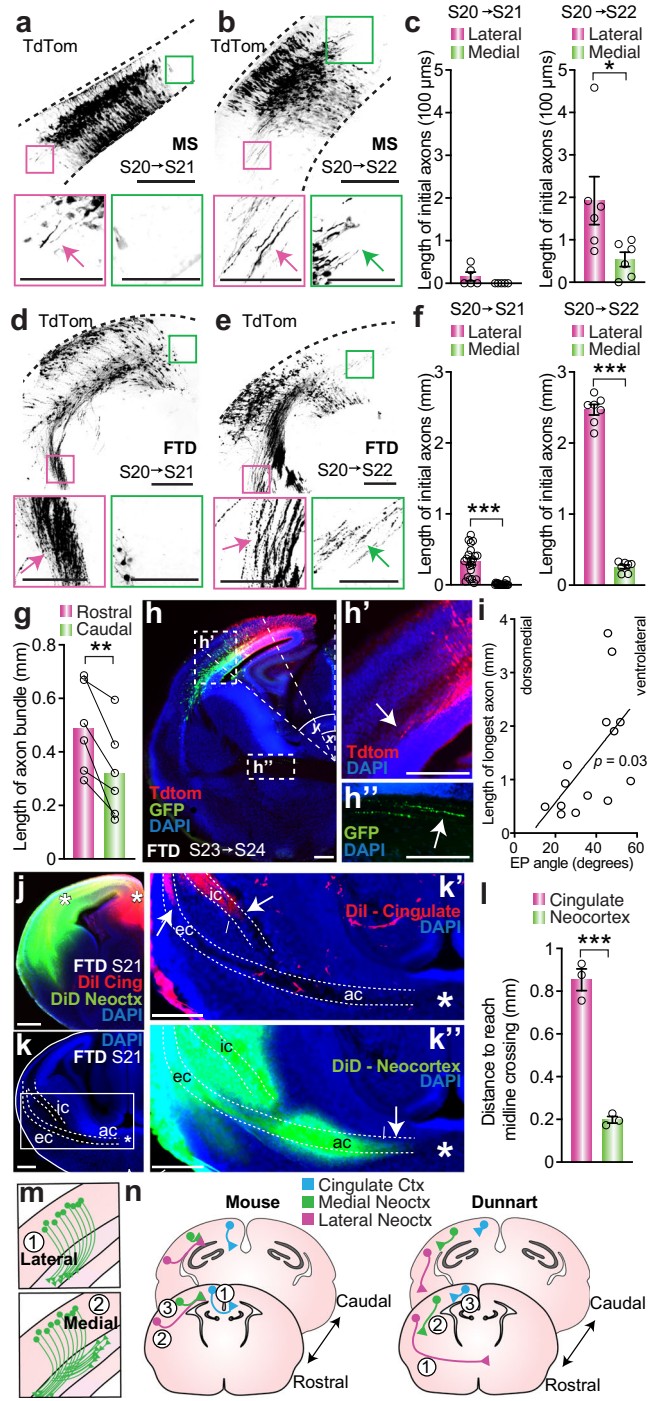

**Fig. 6 | Conserved sequence of neocortical axonal extension, but distinct pioneering populations, in mice and dunnarts. a, b** Mouse and dunnart (**d, e**) cortex electroporated at S20 and collected at S21 (**a, d**) or S22 (**b, e**) and insets of axons extending (arrows) from dorsomedial (green) or ventrolateral (pink) edges of the electroporated field. **c, f** Quantification of the length of the initial axons extending in the dorsomedial or ventrolateral direction following electroporation at S20 and collection at S21 or S22 in mice (**c**) and dunnarts (**f**). N for MS S21:10; S22:22, FTD S21:21; S22:7. **g** Paired comparison of the axonal bundle length from two successive rostral and caudal dunnart electroporations at S23 with either CAG-TdTom or CAG-eYFP and collection at S24. N:6. **h** Dunnart cortex collected at S24 following double dorsomedial or ventrolateral electroporation at S23 with CAG-TdTom and CAG-eYFP. Angles were drawn to the center of more dorsomedially (red, x) and more ventrolaterally (green, y) located electroporated fields, the length of the longest axon of each fluorophore was measured (**h′, h″**). **i** Linear regression of longest axon length and the center angle of the electroporated cell field (dorsomedial: left, ventromedial: right). N for (**g, h**):14 electroporated fields, 10 brains. **j** Image of DiI and DiD ex vivo injection sites in the cingulate cortex or neocortex in S21 dunnarts. **k** DAPI-stained section with dotted lines demarcating the ic, ec and ac and insets showing anterograde axon labeling for cingulate cortex (**k′**) or neocortex (**k″**) injection. **l** Quantification of the distance remaining before anterogradely labeled axons reach the midline through the ac for cingulate cortex and neocortex DiI/DiD injections. N:3 for both. **m, n** Schematic summaries of findings. Data are presented as mean values ±SEM and were compared with Mann–Whitney *U* tests (**c, f, l**), paired *t*-test (**g**) or linear regression (**i**); *p < 0.05, **p < 0.01, ***p < 0.001. See Supplementary Table 3 for exact *p* values and statistical test details. ac anterior commissure, Ctx cortex, ec external capsule, EP electroporation, FTD fat-tailed dunnart, GFP green fluorescent protein, ic internal capsule, MS mouse, Neoctx neocortex, S stage. Scale bars: **a, b, d, e** = 200 μm in the main pictures, 100 μm for insets; **h, h′, h″, j, k, k′, k″** 200 μm.

To better understand this phenomenon, we also compared axon length by days post-electroporation, regardless of overall stage equivalence (Fig. 7k–o and Supplementary Fig. 7f–j). This revealed a more advanced progression of medial axons in mice for both S20 (Fig. 7k) and S23 (Fig. 7n) electroporations, whereas the axon extension comparisons for commissural projections normalized by cortical width were more similar between species (Fig. 7m, o), with dunnarts even overtaking mice at day 3 and day 7 in S23 electroporations (Fig. 7o). This trend was also similar for non-normalized absolute measurements of axon length in S23 electroporated dunnarts (Supplementary Fig. 7j), with no differences being observed between species in this case. These data reveal an overall trend whereby axon extension of the medial vs medial, lateral vs lateral and commissural vs commissural projections may have fixed rates in absolute time, being broadly comparable between species in absolute days. However, due to the relatively protracted development of dunnarts, this results in a remarkably more advanced axon extension in dunnarts compared to mice relative to developmental stage.

### Axonal waiting period in marsupial contralateral targeting

In the final stages of cortical development, axons must locate and innervate their target structures. In placental cortico-cortical commissural projections, this involves a dynamic process of exploration, stabilization and pruning[55]. However, the conservation of contralateral targeting dynamics during development in marsupials has never been explored. To address this, we took advantage of the accessibility of dunnarts developing in the pouch to label DL and UL neurons with S20 and S23 electroporations of different fluorophores within the same cortical hemisphere, and visualize the dynamics of contralateral targeting (Fig. 8a, b). We found that DL neuron axons reached the contralateral hemisphere before UL neuron axons, as expected by their earlier birthdate and maturation (Fig. 8c–f). However, these axons exhibited a waiting period in the white matter underlying the CP, extending only sparse axons into the CP for two stages (around 6 days), until the axons of the UL neurons arrived into the contralateral white matter at S27 (Fig. 8g, h). The two axonal

the main milestones of axonal growth (Fig. 7a–e). Our results revealed that the S20-electroporated medial axons of dunnarts were initially more advanced than those of mice at S22 and S23, with the length of the mouse axons only overtaking that of dunnarts after midline crossing of the corpus callosum (Fig. 7f). Dunnart axons were significantly more advanced by stage for both the lateral versus lateral comparison (Fig. 7g), and commissural versus commissural comparison (i.e. medial in mice versus lateral in dunnarts), with dunnart axons crossing the midline prior to mouse commissural axons (Fig. 7h). These trends were similar for the S23 electroporations (Fig. 7i, j; lateral vs lateral comparisons are not possible in this case as mouse UL neurons do not exhibit any lateral projections), as well as for the absolute measurements of the axons not normalized by cortical width (Supplementary Fig. 7a–e).

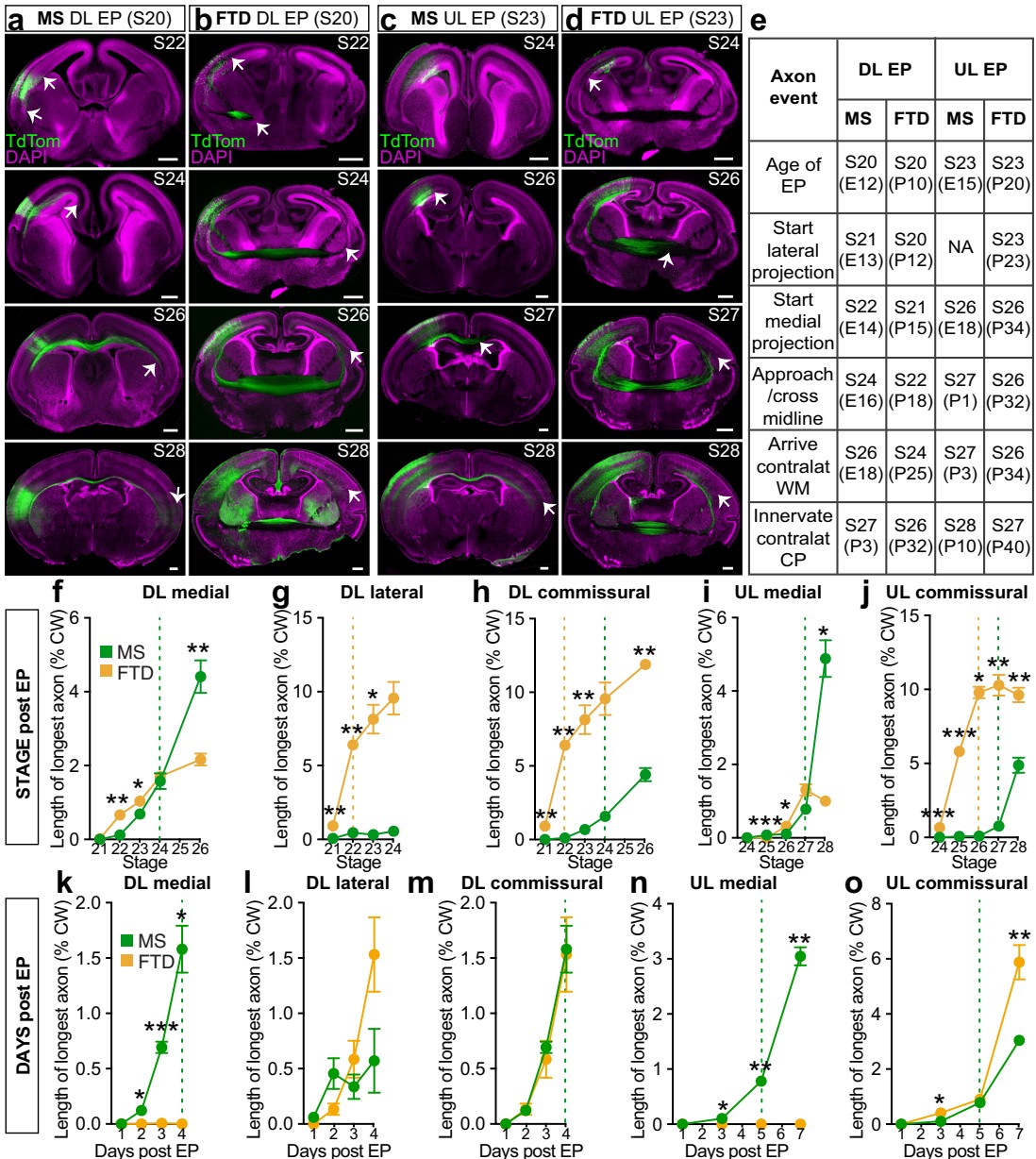

**Fig. 7 | Axon extension is faster in dunnarts than mice relative to developmental stage, but similar in absolute days. a–d** Mice (**a**, **c**) and dunnarts (**b**, **d**) electroporated at S20 (**a**, **b**) or S23 (**c**, **d**) and collected at subsequent stages. Arrows demarcate axon growth. **e** Stages at which milestones of axon growth and guidance occur. **f–j** Quantifications relative to developmental stage of collection of the length of the longest axon extending medially or laterally from S20 (DL) electroporation (**f–g**) or S23 UL electroporation (**i–j**), normalized by the cortical width. Commissural comparisons comprise lateral projections in dunnarts versus medial projections in mice. Stage 23 UL lateral comparison is not possible as axons do not extend laterally from the ULs in mice. Dotted lines represent the stage of midline crossing for mice (green) or dunnarts (orange) N for MS in (**f–h**) S21:5, S22:6, S23:10, S24:3, S26:6; FTD in (**f–h**) S21:21, S22:7, S23:3, S24:4, S26:6. For MS in (**i**, **j**) S24:9, S25:7, S26:5, S27:5, S28:5; FTD in (**i**, **j**): S24:8, S25:13, S26:4, S27:6, S28:5. (**k–o**)

Quantifications relative to the day of collection post-electroporation of the length of the longest axon extending medially or laterally from S20 (DL) electroporation (**f**, **g**) or S23 UL electroporation (**i**, **j**) normalized by the cortical width. Dotted lines represent the stage of midline crossing (where relevant to axonal population) for mice (green) or dunnarts (orange). N for MS in (**k–m**) Day1:5, Day2:6, Day3:10, Day4:3; FTD in (**k–m**) Day1:7, Day2:7, Day3:10, Day4:4. For MS in (**n**, **o**) Day1:9, Day3:5, Day5:5, Day7:9; FTD in (**n**, **o**) Day1:3, Day3:5, Day5:4, Day7:6. Data are presented as mean values ±SEM. All data were compared with pairwise Mann-Whitney *U* tests; *$p < 0.05$, **$p < 0.01$, ***$p < 0.001$. See Supplementary Table 3 for exact *p* values and statistical test details. CW cortical width, DL deep layers, E embryonic day, EP electroporation, FTD fat-tailed dunnart, MS mouse, NA not applicable, P postnatal day, S stage, UL upper layers. Scale bars: 400 μm.

populations then innervated the CP together from S28 (Fig. 8i, j and summary Fig. 8k).

A waiting period of neocortical commissural axons in the contralateral white matter/cortical subplate has been reported for all neocortical layers combined in rats[56] and cats[57], as well as for UL axons in mice[58]. This pause prior to innervation has been hypothesized to be a key period during which innervating axons receive

signals from the overlying cortex and other interconnecting regions and/or acquire certain aspects of maturity. We have previously shown that the adult pattern of interhemispheric connectivity in marsupials shares many features with that of eutherians, such as homotopic targeting and specific hubs of connectivity including the cingulate cortex and claustrum[12]. It is therefore possible that the waiting period between DL and UL projections that we describe here

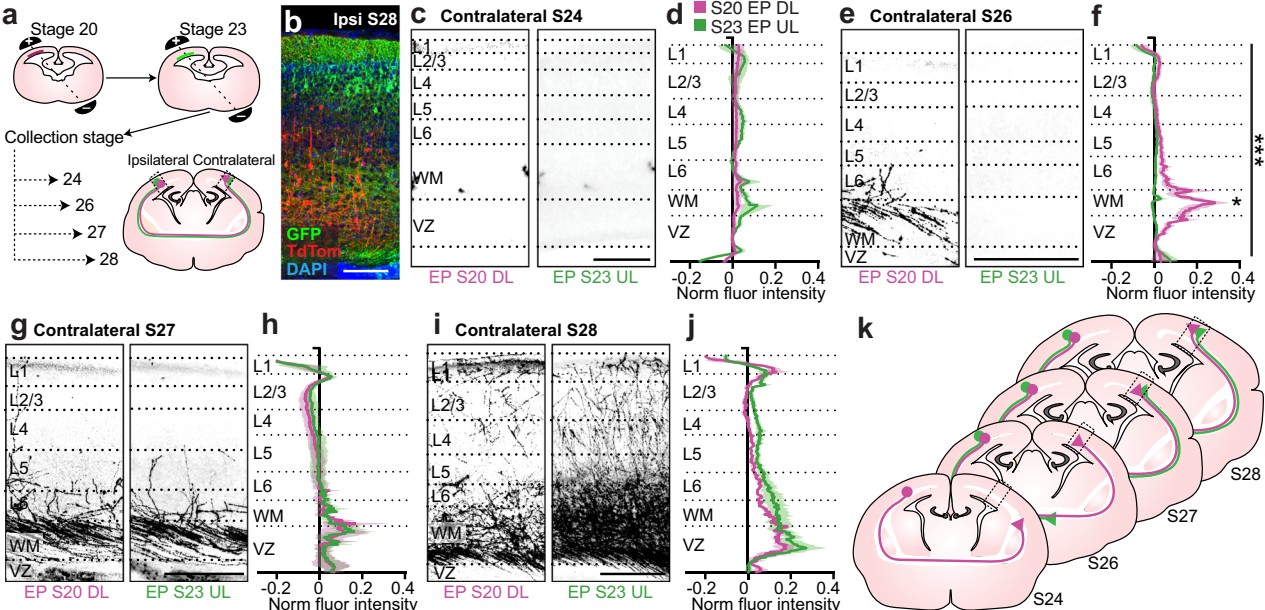

**Fig. 8 | During contralateral targeting in dunnarts, deep layer axons wait for the arrival of the upper layer axons before innervating the contralateral cortical plate. a** Schematic of experimental paradigm. **b** Cortical image of electroporated cell bodies from a S20 electroporation with one fluorophore (TdTom) and S23 electroporation with a second fluorophore (GFP) in the same hemisphere. The brain was collected at S28. **c–j** Images (**c, e, g, i**) of dunnart cortex contralateral to the hemisphere electroporated with different fluorophores at S20 and again at S23 and collected at subsequent stages. Left (pink) is the fluorophore corresponding to the S20 (DL) electroporation and right (green) is the fluorophore corresponding to S23 (UL) electroporation. The average fluorescence intensity of the relevant fluorophore was quantified across the cortical width of the contralateral hemisphere and the average for multiple animals was plotted, with the averaged value of the middle 20% of each cortical layer being quantified and compared between S20 DL and S23 UL electroporation conditions (**d, f, h, j**). N for S24, DL:3, UL:3; for S26, DL:7, UL:5; for S27, DL:3, UL:3; for S28, DL:3, UL:6. **k** Schematic summarizing findings from **c–j**. Data are presented as mean values ±SEM. Two-way ANOVAs or non-parametric omnibus tests were used followed by pairwise $t$-tests where relevant; *$p < 0.05$, **$p < 0.01$, ***$p < 0.001$. See Supplementary Table 3 for exact $p$ values and statistical test details. DL deep layers, EP electroporation, GFP green fluorescent protein, L layer, S stage, UL upper layers, VZ ventricular zone, WM white matter. Scale bars: **c** = 200 μm; **d, f, h, j** = 250 μm.

is also present in eutherians and contributes to some of these shared ancestral features.

### Non-uniform scaling of developmental processes across species

To compare the overall rate of the three processes of cortical development that have been the focus of this study (cell birth/specification, cell migration and axon extension), we plotted rate estimates for each process relative to equivalent developmental stages on a radar plot (Fig. 9a) and represented the findings as fold-change comparisons between species (Fig. 9b). This revealed a similar rate of cell birth/specification between species, whereas migration was modestly faster in dunnarts than in mice. Axon extension comparisons between lateral and commissural axons indicated a much higher rate in dunnarts than mice for all except the medial versus medial comparison of S20 electroporation (DL), which appeared equivalent between species. Taken together, these findings suggest that the successive developmental processes do not scale uniformly, tending to be progressively faster in dunnart than mouse as development proceeds (Fig. 9c).

### Discussion

In this study we report the non-uniform temporal scaling of cell birth/specification, migration and axonal extension in the cortex of mice and dunnarts, two similarly sized species with different developmental timescales. We previously established an equivalent staging system, based on body and head morphological timepoints, to equate developmental milestones across species. Here, we have shown that the timing of cell birth/specification during cortical neurogenesis corresponds with the common staging system in both species; however, the processes of cell migration and axon extension are relatively more accelerated in dunnarts. Therefore, there are important differences in

the relative timing of developmental processes between species that are not solely explained by scalable differences in the staging systems. For example, although the birth of the UL neurons of the cortex occurs at S23 in both species, this takes place in different cortical contexts, i.e. with DL axons already crossing the interhemispheric midline in dunnarts, whereas mouse DL commissural axons have only just begun to extend from the electroporated cell field (Fig. 7b, h). This mismatch could have developmental consequences, such as altered signaling between cell populations that prompt the timing and/or dynamics of subsequent stages of development.

One intriguing hypothetical scenario is that yet unknown cues in the cellular environment (e.g. axon guidance ligands) that prompt the switch between lateral and medial axon projection are linked to developmental stages, for example being controlled by the timing of cell birth/specification. Our data indicate that the timing of this switch is conserved in dunnarts and mice, occurring at around S22 for DL electroporated cells (Fig. 6a–f). However, perhaps due to their advanced maturation/migration, dunnart DL commissural axons begin projecting much earlier than this (S20 in dunnarts versus S22 in mice). This heterochrony may explain the different directionality of commissural axons between species, as dunnart commissural neurons could begin to extend their axons during the "lateral projection" window of environmental cues, before the conserved switch in directionality occurs at S22. A recent study in mice supports this hypothesis, showing that neurons born at an identical timepoint can project to either ipsilateral medial (primary motor cortex) or lateral (secondary somatosensory cortex) targets, and that the major distinguishing feature of these populations is their rate of maturation, with the medial projecting population maturing more slowly[48]. This notion is consistent with our previous findings that precocious expression of the

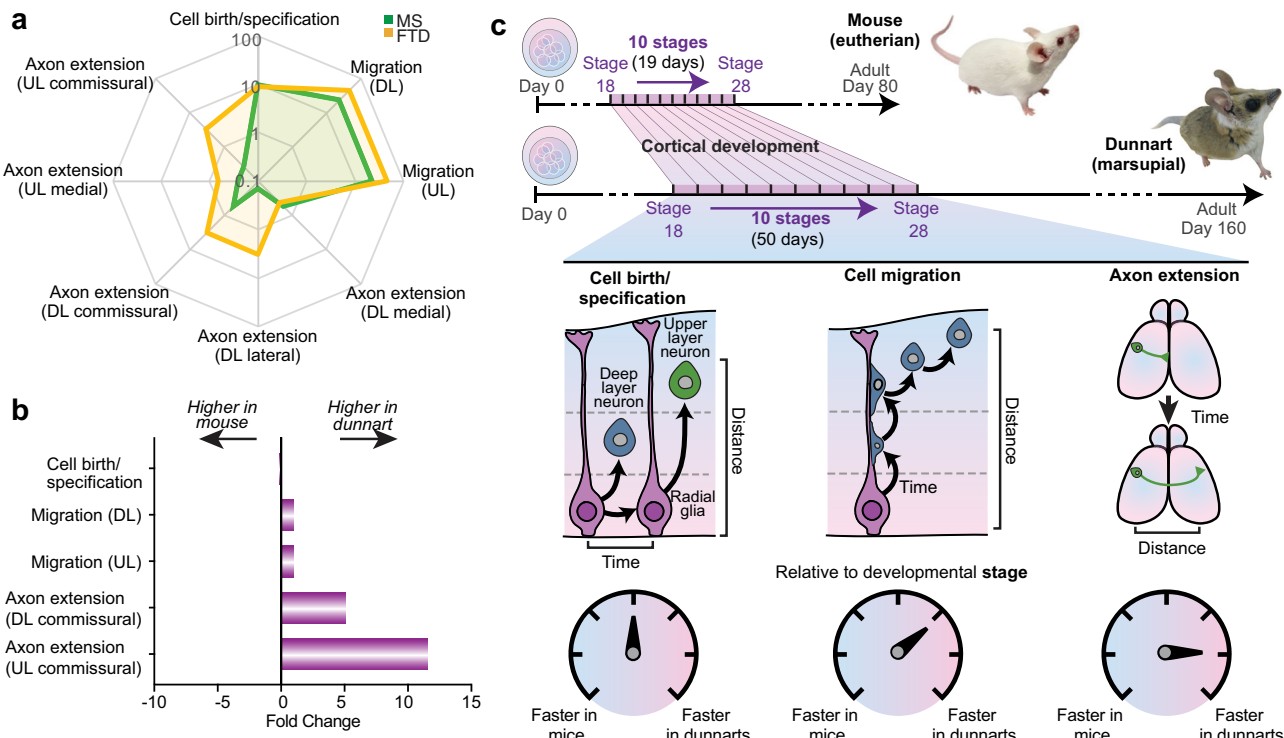

**Fig. 9 | Summary of rate estimate for different cortical development processes. a** Radar plot showing the estimates of rate for each process relative to developmental stage in dunnarts (yellow) and mice (green). The radial scale is expressed as $Log_{10}$ and data used to calculate rate estimates are as follows: cell birth/specification Fig. 1g; migration (DL) Fig. 2b; migration (UL) Fig. 2d; axon extension (DL medial) Fig. 7f; axon extension (DL lateral) Fig. 7g; axon extension (DL commissural) Fig. 7h; axon extension (UL medial) Fig. 7i; axon extension (UL commissural) Fig. 7j. **b** Fold change of rate estimates between species for different cortical processes. Rate estimates were calculated as in (**a**) and fold change between species was calculated for a relevant subset. **c** Summary schematic of non-uniform scaling of timing of developmental processes between mouse and dunnart. DL deep layers, FTD fat-tailed dunnart, MS mouse, UL upper layers.

post-mitotic transcription factor SATB2 promotes the ectopic extension of axons through the anterior commissure in mice, possibly by accelerating the maturation of these neurons to extend an initial axon during the "lateral projection" window. Future studies identifying the cues that mediate this switch, as well as manipulating the timing of neuronal maturation, will help to elucidate whether these phenomena contribute to the divergence of the commissural axon route between species.

A recent study reported that direct neurogenesis contributes to approximately 31.8% of DL neurons and 11.8% of UL neurons in the mouse cortex[59]. If the lack of basal intermediate progenitor cells in dunnarts was the sole explanation for their faster rate of maturation by stage[39], then we would expect the subpopulation of mouse neurons produced via direct neurogenesis to match the rate of dunnarts. However, this was not the case, as our analysis of the most advanced 2% of cells (which are likely to be produced via direct neurogenesis in both species) also revealed less advanced maturation in mice compared with dunnarts at the earliest stages (Supplementary Fig. 2c, f). As our axon analysis was based upon the longest axon, it also would have encapsulated the fastest axons in both species (Fig. 7). Therefore, although the absence of basal intermediate progenitor cells may explain some of the differences in maturation rate between species, other mechanisms are likely to contribute to this divergence. One possible explanation for this differential temporal scaling is that the timing of cell cycle and cell birth is very plastic to change, as has previously been shown both within species over time and between species[6,60,61]. Relatively less is known about the temporal plasticity of cell migration and axon extension, with a handful of studies indicating that different species may have distinct rates for these processes[62–64]. However it remains unclear whether this is the case in vivo, and to what extent differences are due to changes in the rate of cell/axon

movement versus pauses/waiting periods[65]. One interpretation from this study is that cell migration and axon extension may be more fixed in their absolute rate than cell cycle, with the result that they are relatively faster according to developmental stage in slower developing species. Future work using live imaging across many species will be required to address these dynamics in detail.

An important aspect to consider when comparing time scales between marsupials and eutherians is the context in which each of them develops (i.e. cortical neurogenesis occurs entirely postnatally in dunnarts versus entirely prenatally in mice). This difference raises two questions. First, to what extent is the relatively faster rate of migration/axon extension in dunnarts necessary to facilitate early behaviors concomitant with ex utero development? There has been debate over the nature and degree of sensory input (olfactory, taste, thermal, somatosensory and gravitational) and motor output[66–68] that is present in marsupials at these early stages, as well as the level of the central nervous system that controls these putative functions[69,70]. Despite this, it is likely that dunnart joeys receive more sensory stimulation than mice at an equivalent stage in utero, and also display more active and directed behaviors as they navigate to the teats, possibly reflecting higher selective pressures for more mature brain circuits. However, the timing of cortical development in the common ancestor of Theria remains unknown, as well as how temporal differences might have originated between lineages.

The second question to consider is the extent to which the differing developmental environments of dunnarts and mice affect their developmental timing. The aforementioned sensorimotor functions that occur postnatally in marsupials may not only require advanced neuronal maturation, but could also permit and/or instruct aspects of neurodevelopment, as been shown for multiple sensory systems in postnatal eutherian species[71,72]. The early birth of marsupials is also

likely to result in many physiological differences between species at these early stages. For instance, small marsupials respire solely through their skin for the first few postnatal days[73,74] and have a lower rate of total oxygen consumption than predicted based on body mass compared to newborn placental mammals[75]. However, it is difficult to directly compare or predict the oxygenation of progenitor cells in marsupial pouch young compared to the mildly hypoxic in utero environment of placentals[76], where fetuses exchange all gases via the umbilical cord that delivers maternal blood supply, especially considering the effects of other reported differing systemic factors in marsupials such as hemoglobin properties[77,78] and vascular structure[79]. Nevertheless, evidence from mice has shown that ectopic alterations in oxygen, blood supply and metabolic processes can influence the dynamics of cortical neurogenesis, including the reduced proliferation of basal intermediate progenitor cells in response to experimentally reduced gestational oxygen[80–83]. An intriguing scenario is that differences in reproductive strategy, that confer, for example, lower brain oxygenation in dunnarts, directly regulate aspects of neurogenesis, such as basal intermediate progenitor cell proliferation, and ultimately developmental tempo.

Subtle differences in the timing of distinct cortical development processes could lead to evolutionary innovations if they persist throughout generations, highlighting the modular evolvability of cortical features. Further research into the phenotypic consequences of varying developmental timing and environment within and between species will help to address these questions, as well as continuing to expand our understanding of the temporal constraints and plasticity of ontogenetic events.

## Methods

### Animals and tissue collection

All animal procedures, including breeding, were approved by The University of Queensland Animal Ethics Committee and the Queensland Government Department of Environment and Science and were performed according to the current Australian Code of Practice for the Care and Use of Animals for Scientific Purposes (NHMRC, 8th edition, 2013). CD1 mice were time-mated, with the day of separation considered embryonic day 0. Fat-tailed dunnarts (*Sminthopsis crassicaudata*) were time-mated and pouch checked at least three times per week, with the day of identified pouch young designated as postnatal day 1. Both mice and dunnarts were housed in an animal facility under standard 12 h/12 h light/dark cycles with food and water ad libitum, 18-24 degrees Celsius and 30-70% relative humidity. As sex was not identifiable in such young embryos/pouch young, data were not disaggregated according to sex.

Deep anesthesia of mice for recoverable procedures (i.e. in utero electroporation) was obtained with an intraperitoneal injection of ketamine/xylazine (120 mg/kg ketamine and 10 mg/kg xylazine). For sedation of adult female dunnarts with joeys, animals were transferred into a gas anesthesia induction chamber with 5% isoflurane, delivered in oxygen at a flow rate of 200 ml/kg/min. The anesthesia was then maintained by supplying 2–5% isoflurane through a silicone mask (Zero Dead Space MINI Qube Anaesthetic System) throughout the procedure. For terminal collection, the joeys were removed from the teat by gently pulling with forceps and anesthetized with an intraperitoneal injection of 190 mg/kg sodium pentobarbitone, or 5-10 min ice anesthesia for joeys younger than S28 (less than P40). Adult dunnarts and mice, as well as mouse pups, were deeply anesthetized with an intraperitoneal injection of sodium pentobarbitone. Joeys and mouse pups younger than S21 (P15 and E13, respectively) were decapitated and drop fixed in 4% (w/v) paraformaldehyde (PFA) in 1X phosphate-buffered saline (PBS), whereas older joeys and mouse pups, as well as adults, were transcardially perfused with 0.9% (w/v) saline followed by 4% (w/v) PFA. The brains were subsequently post-fixed in 4% (w/v) PFA for four days prior to processing.

## Animal procedures

**Intraperitoneal injections of EdU in mice and dunnarts.** Pregnant mouse dams were injected intraperitoneally with a thymidine analog, ethynyl deoxyuridine (EdU, 5 µg/g of body weight, diluted in water) via a 27 G needle at different stages of mouse embryonic development (from S18, E10 until S25, E17; see Supplementary Fig. 1a for staging system). Female dunnarts with pouch young were anesthetized as described above; the dunnart joeys were exposed carefully and injected intraperitoneally using a glass-pulled pipette attached to a picospritzer II (Parker Hannifin) at different postnatal stages (from stage 18, P0- P3 until stage 25, P27-30) with EdU (5 µg/g of body weight, diluted in water). The mouse pups and the joeys were then euthanized and transcardially perfused at S27 (P36-P40 for dunnart joeys, P4-P5 for mouse pups), with the tissue being processed for the detection of EdU using the Click-iT EdU Imaging Assay. The brains of the mice and dunnarts injected with EdU were the same as those used in previous studies focusing on different questions[16,26]. Specifically, in[16], EdU-labeled neocortices were used to birthdate SATB2+ and CTIP2+ cells in mice and dunnarts, whereas in[26], they were used to generate microscopy images that qualitatively showed the timing of neocortical layer generation (no quantification was performed in this publication).

**In utero electroporation of mice.** Time-mated CD1 pregnant dams were used at S20 (E12, deep layers neurogenesis in the neocortex) or S23 (E15, upper layers neurogenesis) for all experiments. Deep anesthesia was achieved as described above. Following exposure of the uterine horns via laparotomy, various combinations of plasmids (Supplementary Table 1) were microinjected in the lateral ventricle with a picospritzer II holding a glass pulled pipette. The plasmids were then electroporated into the right primary somatosensory cortex (S1) with 3 mm diameter microelectrodes (Nepagene) delivering 5 (100 ms, 1 Hz) approximately 36 V square wave pulses from an ECM 830 electroporator (BTX Harvard Apparatus). Once this procedure was completed for each embryo, the uterine horns were replaced inside the abdominal cavity and the incision was sutured closed. Animals were then subcutaneously injected with 1 ml of Ringer's solution and recovered in a humidified chamber at 28 °C. For analgesia, buprenorphine (0.05 mg/kg) and meloxicam (5 mg/kg) were injected subcutaneously, and edible buprenorphine was injected into jelly and placed in the cage.

**In pouch electroporation of fat-tailed dunnarts.** Once the female dunnart was anesthetized as described above, the pouch was carefully everted, and the joeys were exposed gently. The lateral ventricle of the joeys was injected with the appropriate plasmid or combination of plasmids (Supplementary Table 1) using a Picospritzer II holding a glass-pulled pipette, after which 1 mm forcep-type electrodes (Nepagene) were positioned and used to deliver five 100 ms square pulses of 30-35 V via an ECM 830 electroporator.

**Plasmids.** Plasmid concentrations and sources are provided in Supplementary Table 1. Plasmids were prepared for electroporation with a QIAGEN Endotoxin-Free Maxi Prep kit as per the manufacturer's instructions. All plasmids were combined in sterile PBS with the addition of 0.0025% Fast Green dye to aid visualization.

**Ex vivo retrograde tracing following DiI/DiD injection.** The brains from four P12 dunnarts were collected, drop-fixed in 4% PFA and subsequently dissected out of the skull. Each brain was secured to a plastic dish using a small amount of agar on the ventral surface only. Carbocyanine 1,1′-dioctadecyl-3,3,3′,3′-tetramethylindodicarbocyanine perchlorate (DiI) or 1,1′-dioctadecyl-3,3,3′,3′-tetramethylindocarbocyanine 4-chlorobenzensulfonate (DiD) was applied to the tip of a glass-pulled pipette and mounted in a micromanipulator. Using a 5x objective microscope, the manipulator was advanced to place a crystal of DiI or

DiD into the cortical plate of either the somatosensory cortex or the cingulate cortex, with care taken to limit the depth of advancement in order to avoid the developing white matter tracts. The tracers were used alternately for each site in each animal. The brains were then placed in 2% PFA and incubated at 37 °C for 2 weeks, followed by incubation at room temperature for a further 3 weeks. They were then sectioned at 50 μm on a vibratome and each brain was mounted en toto on a slide. A counterstain of 0.1% 4′,6-diamidine-2′-phenylindole dihydrochloride (DAPI), was applied for 10 min, after which the slide was washed and coverslipped with ProLong Gold antifade mounting medium.

### Histology and image acquisition

**Brain sectioning.** Mouse and dunnart brains were dissected out of the skull, embedded in 3.4% agarose, glued to a magnetic specimen disk, positioned in a tray filled with 1X PBS and sectioned in 50 μm coronal slices using a vibratome (VT1000S, Leica Biosystem,). A representative section from each brain region, from rostral to caudal, was mounted onto each slide.

**Fluorescence immunohistochemistry.** Slide-mounted brain sections were fully dried and where required by the antibody-antigen combination, incubated in 4% PFA for 10 min. Antigen retrieval was performed in a decloaking chamber (DC2012, BioCare) with 0.01 M sodium citrate (pH 6) with 0.05% (v/v) Tween-20 for 4 min at 110 °C and 15 psi. Epitope blocking was achieved by incubating sections in 1X PBS (pH 7.4) with 10% (v/v) normal donkey serum (NDS) and 0.2% (v/v) Triton X-100 (TX100). Blocking was followed by an overnight, room temperature incubation with primary antibodies (Supplementary Table 2) in 1X PBS (pH 7.4) with 10% NDS and 0.2% TX100. After three 1X PBS washes for 20 min, slides were incubated with the appropriate fluorescent secondary antibody (see Supplementary Table 2) in 1X PBS with 0.2% TX100. A streptavidin-based amplification was used for some of the primary antibodies to optimize labeling. After an additional three 1X PBS washes, the slides were incubated for 10 min in 1X PBS with 0.1% (v/v) DAPI, then washed and coverslipped with ProLong Gold AntiFade mounting medium. 50 μm brain sections from dunnart joeys and mouse pups injected with EdU were processed according to the instructions of the Click-iT EdU imaging kit before being incubated with the blocking solution. For some difficult to label antigens or antibodies which produced high background fluorescence, 200 mM glycine and 0.5% (v/v) bovine serum albumin (BSA) were added to the blocking buffer, and sections underwent additional incubations in 1X PBS with 0.25% (v/v) BSA, 200 mM glycine and 0.2% (v/v) TX100 following PBS wash steps and prior to antibody incubations.

**Image acquisition.** Wide-field fluorescence imaging was performed with a Zeiss upright Axio-Imager Z1 microscope fitted with Axio-Cam HRc and HRm cameras and captured with Axio Vision software using MosaicZ. Images were acquired with Zen software (Carl Zeiss). High resolution images were acquired using 1) a CFI Plan Apochromat VC 20x/0.75 NA air objective (Nikon) or CFI Apo Lambda S LWD 40x/1.15 NA water immersion objective (Nikon) on a Diskovery spinning disk confocal microscope (Spectral Applied Research) built around a Nikon TiE body and equipped with two sCMOS cameras (Andor Zyla 4.2, 2048 ×2048 pixels) and controlled by Nikon NIS software 2) a spinning-disk confocal system (Marianas; 3I), consisting of an Axio Observer Z1 (Carl Zeiss) equipped with a CSU-W1 spinning-disk head (Yokogawa), ORCA-Flash4.0 v2 sCMOS camera (Hamamatsu Photonics) and 20 × 0.8 NA PlanApo objectives. Image acquisition was performed using SlideBook 6.0 (3I) or 3) a Zeiss LSM900 inverted laser scanning confocal microscope with AiryScan2 GaAsP-PMT detector and a 20x/0.80 NA Plan-Apo air objective. Images from the LSM900 were acquired using SR-2Y multiplexing, with deconvolution and pixel reassignment applied via

Airyscan post-processing in the Zen Blue software (Carl Zeiss). Images were cropped, sized, and enhanced for contrast/brightness with ImageJ and Photoshop, and the figures assembled in Illustrator (Adobe Creative Cloud).

### Analysis

**Cell counting.** Cell counting analyses in specific regions of interest (ROIs) were performed manually with a Fiji (Image J) plugin for manual cell counts. Differences between species were observed in both absolute cortical thickness and relative layer thickness. To facilitate between-species comparisons, EdU-positive cell positions were mapped onto a common axis. This was achieved by first averaging the layer marker positions within each species, and normalizing cell positions as a percentage of cortical width. Although this normalization accounted for differences in absolute thickness, it did not allow for differences in relative layer thickness. To achieve this, layer positions between species were averaged, creating an "idealized" common axis by which mouse and dunnart cell positions were mapped to using layer-specific scaling factors. After this correction, the median EdU position was calculated and compared between species. For "leading edge" quantifications, the top 98th percentile of the distribution of counted cells was calculated using the "Percentile" function in Microsoft Excel, with the result expressed as % of cortical width by dividing by the total cortical width for that ROI. Violin plots (Fig. 3c and 5g, and Supplementary Figs. 3e, n, p and 5a) were constructed by transforming the y-coordinates of each TdTom+ cell (or PH3+ or TdTom+PH3+ cell for Fig. 5g and Supplementary Fig. 5a) in every animal to a percentage value relative to the entire width of that animal's cortex. Each of these values was then transformed onto the average cortical width (in μm) of all animals per "days post-electroporation" group and the entire distribution for every counted cell in every animal in that group was plotted as a single violin plot in GraphPad Prism. Dotted lines represent median and quartiles. Bands underlying the violin plots demarcate the averaged location of the borders revealed by relevant immunostaining for that cohort of animals.

**Layer demarcation.** The position of cortical layers and zones was demarcated using either DAPI labeling and/or labeling with classic cell markers as indicated. In all cases, the position of cortical layers and zones was informed by prior immunohistochemical stains of classic cell markers in this article as well as others published by us[16,26]. Differences in the age of collection and/or the coexpression of markers resulted in some differences in how the cortical layers/zones were separated/combined throughout the manuscript (although these were always the same in a single between-species comparison), and depended on the confidence with which we could accurately and consistently separate partitions (e.g., separation versus combination of layers 5 and 6). The adult mouse and dunnart brain sections labeled with SATB2 and CTIP2 (Fig. 1j, k) were the same as those used in one of our previous papers[16]. However, the quantification performed in[16] was different from the analysis in this current study. Specifically, here we used SATB2 and CTIP2 to demarcate neocortical layers, while in[16] (Fig. 2) we counted cells labeled with SATB2 and CTIP2 in each cortical layer in both mouse and dunnart.

**Choice of ROIs.** To ensure that the ROIs cropped from the original microscopy images were an adequate representation of the neocortical region being analyzed, their width spanned from 50 μm to 300 μm, depending on the experimental paradigm and the type of analysis being performed. ROIs were taken from the primary somatosensory cortex unless otherwise specified.

**Axon analysis.** Axon length was measured manually using Fiji plugins. For the analysis of axonal contralateral targeting, a ROI was defined in the hemisphere contralateral to the electroporated patch and analyzed

using Fiji Plot Profile. The intensity of the fluorescence profile in these ROIs was normalized by subtracting the fluorescence intensity measured in the darkest brain region (contralateral striatum; background), divided by the brightest brain region (ipsilateral white matter, controlling for brightness of the electroporated patch). For statistical comparison, the averaged value of the middle 20% of each cortical layer was calculated for each individual and compared between conditions.

**Calculation of rate estimates.** Rate estimates (Fig. 9) were calculated by subtracting the relevant mean distance value (per species) of the latest stage means that were or could be assumed to be zero (e.g. at the time of electroporation for migration and axon extension) or the earliest measured stage, from the mean distance value (per species) of the stage of the highest normalized mean. This difference in distance was divided by the number of intervening stages (time), ceasing the measurement at midline crossing where relevant. Fold change was calculated by subtracting the mouse rate estimate from the dunnart rate estimate for each relevant measurement and dividing this by the mouse value.

**Statistical analysis.** Individual statistical tests for each comparison as well as sample numbers and outcome of assumptions tests are detailed in Supplementary Table 3. The Python (3.9.15) and R (4.2.0) programming languages were used for data processing and statistical computing. Analysis packages included: *ARTool* (R; 0.11.1), *compositions* (R; 2.0.6), *npmv* (R; 2.4.0), *pingouin* (Python; 0.5.3), *skbio* (Python; 0.5.8), *statsmodels* (Python; 0.13.2), *scipy* (Python; 1.9.3), and *pandas* (Python; 1.5.2). Data were first visualized using boxplots, to determine the presence of putative outliers. The data were then z-scored and absolute values >2.5 were considered potential outlying values and were inspected for biological inconsistencies (i.e. the sample was taken from a discernibly more rostral or caudal part of a brain region). If clear biological inconsistencies were observed, the sample was removed from subsequent analysis. However, if no biological explanation was uncovered (i.e., the sample was highly comparable to the others), then outliers were retained. The assumptions of parametric models were evaluated for all data. The normality of residuals was assessed using the Shapiro-Wilk test, as well as the inspection of Q-Q plots. Deviations from normality were considered significant at the 0.05 alpha level. Homoscedasticity was assessed using Levene's test. Comparisons between two independent groups were analyzed using a *t*-test or Mann–Whitney *U* test, depending on assumption outcomes. In some cases, two groups were paired and analyzed using a paired samples *t*-test, or a Wilcoxon signed-rank test in cases where differences of paired data were non-normal. Where omnibus testing was appropriate, and parametric assumptions were met, data consisting of two between-subjects factors and a single response variable were examined using two-way ANOVA, followed by pairwise *t*-tests. For multiple comparisons, the false discovery rate (FDR) was controlled using the Benjamini-Hochberg procedure (5% FDR). In cases where parametric assumptions were significantly violated, data of this form were instead analyzed using the Aligned Rank Transform method, a non-parametric alternative to factorial ANOVA, implemented in *ARTool*[84,85]. For some comparisons (i.e. all comparisons that were made across multiple time points), overall effects were of less interest than specific age-dependent developmental differences between species. In these cases, omnibus tests were not performed, and data were analyzed directly by pairwise comparisons using either pairwise *t*-tests or Mann-Whitney *U* tests depending on assumption outcomes.

Some analyses involved comparisons of proportions. Due to the unit-sum constraint of compositional data, assumptions of independence could not be made for some comparisons. To account for the compositional nature of these data, proportions were transformed using the isometric log ratio method, using either the *skbio* Python package or *compositions* R package[86,87]. Subsequent testing was performed using multivariate omnibus tests. Assumptions of multivariate normality and equivalence of covariance matrices were evaluated using the Henze-Zirkler (HZ) normality test and Box's M test ($\alpha = 0.001$), respectively. If assumptions were met, data were analyzed by MANOVA; otherwise a non-parametric method for the analysis of multivariate samples, *npmv*, was used[88,89]. Where possible, a Wilks' Lambda test statistic was used; however, if unavailable (i.e., when the rank covariance matrix was singular), an ANOVA-type test statistic was reported instead[88]. In some instances, cortical layer proportions were zero (e.g., TdTomato⁺GFP⁺ in the cortical plate, Fig. 4c). To avoid divide-by-zero errors, a small value (0.0001) was added to the proportion and subtracted evenly among the remaining proportions. If significant differences were detected, then bootstrapped ($n = 5000$) log ratios of group geometric means were computed to generate 95% confidence intervals for each layer[89,90]. Layer differences were considered significant if confidence intervals did not include a log ratio equal to 0. Similar to prior implementations of this approach, exact *p*-values were not computed but instead reported according to estimated confidence intervals. All tests performed were two-sided. All data were visualized using GraphPad Prism, with error bars representing the standard error of the mean. Unless stated otherwise, a significance level of 0.05 was assumed for all tests, with significance graphically represented as $*p < 0.05$, $**p < 0.01$, and $***p < 0.001$. Significant omnibus and pairwise comparisons are demarcated graphically using lines and brackets, respectively. In all cases, N values are the numbers depicted in the graphs and those listed in Supplementary Table 3 are of different individual animals.

### Reporting summary
Further information on research design is available in the Nature Portfolio Reporting Summary linked to this article.

## Data availability
The raw quantification (e.g. cell counts, axon length quantifications etc) data used for statistical tests are provided in the Source Data file. Source data are provided with this paper.

## Code availability
For analysis of compositional data, a custom Python analysis script was written and is available at https://github.com/Fenlon-Suarez-Lab/histology-analyses/tree/main/Paolino-2023.

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

## Acknowledgements

We gratefully acknowledge the staff at the Advanced Microscopy Facility at the Queensland Brain Institute (QBI) and the Core Imaging Facility at the School of Biomedical Sciences (SBMS) the University of Queensland (UQ), for their support and assistance. We thank the UQ

Biological Resources and the Native Wildlife Teaching and Research Facility for help with all animal husbandry. Thanks to R. Tweedale, U. Siebeck, and P. Kozulin for their support during experiment design, data collection, and manuscript preparation. This work was supported by the Australian Research Council (Discovery Early Career Researcher Award DE160101394 to R.S., Discovery Project Grant DP160103958 to L.J.R. and R.S., and a Discovery Project Grant DP200103093 to R.S. and L.R.F.), the National Health and Medical Research Council (Principal Research Fellowship GNT1120615 to L.J.R., Investigator Grant Emerging Leader 1 GNT1175825 to L.R.F, and Ideas Grant 2013349 to R.S.), Ikerbasque - Basque Foundation for Science (F.G-M.), the Brain and Behavior Research Foundation (26728 to R.S. and 30819 to L.F.), The University of Queensland (postgraduate scholarships to A.P., E.H., and E.J.B., Development Fellowship to L.R.F., and Amplify Fellowship to R.S.) and National Institutes of Health P01 grant to L.J.R.

## Author contributions

Contributions: A.P., E.H., E.J.B., F.G-M., L.J.R., R.S., and L.R.F. designed research; A.P., E.H., E.J.B., D.A.B., C.M., R.S., and L.R.F performed research; A.P., E.H., E.J.B., D.A.B., and L.R.F. analyzed data; A.P and L.R.F drafted the manuscript and all authors provided input.

## Competing interests

The authors declare no competing interests.
