## [Peer Review File · Nature Communications]

Non-uniform temporal scaling of developmental processes in the mammalian cortexREVIEWER COMMENTS

Reviewer #1 (Remarks to the Author):

In this paper, Paolini A. and colleagues generate an extensive comparative analysis between FTD (a marsupial) and mouse (an eutherian) cortical development. This work is based on a large and detailed developmental timeline alignment between the two species, allowing developmental stage-based comparison. By combining neuronal birthdating techniques and in utero electroporations, the authors detail the pace of neuronal production, migration and maturation in both species. The study shows that the same developmental processes tend to occur faster in marsupials, suggesting a neoteny in eutherian cortical development and setting the ground for the complexification of neocortical structures. The authors relate this more rapid pace to the absence of intermediate progenitors, a key cell type to produce late-born neurons in eutherian species. All findings are nicely summarized in a final figure.

This study tackles an important evolutionary question and is of broad relevance. Points of discussion and suggestions are presented below:

- In the text, the terms basal progenitors and intermediate basal progenitors are used in an interchangeable manner. The terminology should be streamlined. More importantly, the lack of IPs in the FTD seems to be at the root of all the other observations, yet this point is not well presented in the text, results and discussion. Because of its mechanistic nature, I strongly recommend that the authors emphasize this point more, including in the title.
- Figure 1: It is striking how thin L2/3 appears in panel J. In the somatosensory barrel cortex of the mouse, L2/3 and L4 have roughly the same width. With a DAPI staining it is difficult to appreciate layer limits. In order to quantify or at least display this, it would be better to use layer-specific markers.
- Figure 3: in order to display differences across developmental time points and species, for panel C, it would be necessary to normalize cortical width to 1. From Figure 1i, it is clear that marsupial cortex is thinner in adults. This would also help to better illustrate the evolution of cell-type proportions across development.
- Figure 5: please comply with a homogeneous terminology for intermediate progenitors.
- Figure 5: as for figure 3, please normalize cortical thickness to 1 in order to compare cell-type proportions and distribution across laminae. Although unlikely, the observed bimodal distribution of mouse PH3+ cells could also be observed in marsupials but the representation doesn't allow this to be assessed. A comparable scale could help to reinforce the author's findings.
- Figure 5: single cell RNA seq dataset, are IPs present? Or are Tbr2+ cells exclusively non-cycling?
- Figure 6: It is interesting that between the two species, there is an opposing dynamics in contralateral targeting: medial to lateral in mouse, vs lateral to medial in FTD. It appears, however, difficult to compare the projection dynamics of mouse medial callosal neurons to FTD medial intracortically projecting neurons. Is this intracortically projecting population present in mice? In FTD, this medial projection may be independent of anterior commissure projections. Perhaps reinforcing the discussion on the emergence of medial callosal projecting neurons in mice would be enough.
- Figure 7: the absence of intermediate progenitors can indeed explain the extension of the pace of migration and maturation of indirectly-born neurons in the mouse cortex. However, directly born neurons should exhibit similar developmental tempos. This could be assessed by comparing the fates of electroporated neurons that remain unlabeled following chronic BrdU injections (i.e. born from cells that never underwent a second round of division after electroporation) to those that are labeled. Flashtag could be used instead of electroporation for this purpose too.
- Figure 8: relating to the waiting period of DL contralateral projections, does this promote homotopic projections specificity? In mice there is some degree of areal refinement across postnatal development. Is this also the case of the FTD or it is prevented by the presence of this waiting period?

Reviewer #2 (Remarks to the Author):

The authors are to be congratulated for a much-needed and long-awaited empirical contribution to an understanding of the scaling of developmental processes underlying the generation of the cortex in mammals. The methods are well-chosen, optimal for their purposes, and the quantification is fine. As a general editorial comment, I'd note that their results sections and discussion suffer from the alphabet-soup of acronyms generic to present developmental neurobiology. While they do a good job in their summary titles of each section describing what will be covered, I would suggest injecting more redundancy throughout (for example, the function of a particular tracer, names and significance of layers or developmental compartments) to help the non- or not-yet expert along. Similarly, the figures, constrained by the need to pack 40 into 4, seem more to attend to packing than communication. For example, the graphs which compare timing on a maturational scale for mouse versus dunnart would naturally stack vertically for best comparison; it might even save some space. Now, they appear to have no particular system. Reduction in graphs/figure and more parallel construction would help.

The only substantial criticism I have is with the word "non-uniform", which in the title is applied to temporal scaling of developmental processes. "Scaling" implies a comparison, or some sort of normalization, but just what temporal scaling might be is neither commonly done nor understood very well, in my experience. What about developmental processes could be scaled and what "non-uniform" means in this context might be better spelled out.

For example, a great deal is known about the intrinsic variability of neurogenesis, with respect to its absolute time, – absolute time to generate a neuron is variable across species (and thus sometimes temperature), brain region and cell class; the duration of a cycle of neurogenesis typically extends with maturational stage, but not all cell cycle components are lengthened uniformly, etc. The authors do a good job in dissecting several of the possible components, but are hampered in laying out an explanation of "non-uniform temporal scaling", by the fact that the product of neurogenesis, the cortex, is chosen to be as similar as possible for this investigation. To produce a similar end number of neurons, but over a duration three times as long, what possibilities for alteration of cell cycle kinetics are even possible? Considering absolute time, not only layer-specified targeting of neuron number, what has scaled to produce such uniformity? It would be interesting to compare (not new studies, just extant literature) controls for absolute time, not absolute neuron output – comparisons to the eutherian ferret or cat cortex (or retina). Relative and absolute extensions of time might be informative, as in the Cebus versus Aotus retina (Dyer et al., PNAS '09).

When it comes to neuronal migration and axon extension, comparatively very much less is known about absolute rates of progress, by species or maturational stage. Both developmental phenomena would seem to have fewer "targets" for modification (i.e. base stem cell number, the duration of the phases of the cell cycle, and alteration of the number of cells cycling, and more!). The authors' results suggest that in fact, neither cell migration nor axon extension reflect species-of-origin (could the rate be constant?), and thus stack up "early" in the maturational state of the slower maturing dunnart. An interesting suggestion is that these "mismatches" are buffered systemically -- by an SVZ as an outflow region for cases where neurogenesis exceeds migration capacity (Charvet & Striedter 2011: Causes and consequences of expanded subventricular zones, Eur. J. Neurosci), or by a "waiting period" for tardy migration as shown here, and in citations. The authors might be presenting a central case of "phenotypic integration" or "evolvability", the present paper a first step in building such a systemic understanding. The unlikelihood of vast accelerations in the pace of such axon outgrowth, given the "rodent" (both types) to monkey range should be presented to make clear what "non-uniform" would mean!

Overall, while I appreciate they are trying to present a consistent definition of a maturational-stage based temporal scaling, failing to provide a concrete anchor of just what might be scalable in a

developmental mechanism would help comprehension. Providing it would give the paper more impact.

Reviewer #3 (Remarks to the Author):

The paper by Fenlon et al provides very unique insights into Understanding the rules and limitations of timing in neocortical development and how this is mediated by changes in gene expression. The unique use of a marsupial model and exquisite tools developed in this species to visualise these dynamics is commendable and shed new insights into the development and evolution of the mammalian brain.

This is an extremely careful anatomical study – rarely seen with such precision. Novel insights into heterochrony in brain development between mice and dunnarts. The paper flows on from a careful staging series of the dunnart head which describes and aligns developmental milestones in neurogenesis with the mouse model. This underpins the work here, showing the differential emergence and migratory paths of marsupial neurons relative to the developmental stage.

The study is well described, the data are beautifully and clearly presented. And the outcomes are clear. One of the only points I had was surrounding the rationale for these differences. In the discussion the authors make the statement 'cortical neurogenesis occurs entirely postnatally in dunnarts versus entirely prenatally in mice'... 'Despite this, it is likely that dunnart joeys receive more sensory stimulation than mice at equivalent stage in utero'

Presumably olfactory and gravitational forces are important for navigation to the pouch ie. before birth, do these early events drive changes in early brain progenitors? Once in the pouch, external stimulation is somewhat muted and the only stimuli they need to respond to is suckling? I wonder if the authors can comment on / expand the uniqueness of this requirement, the marsupial brain at birth and this might underpin some of the findings here?

The oxygen scenario is also an interesting one. Can the authors state if they are suggesting the dunnart has a reduced or increased oxygen environment relative to a mouse in utero? What are oxygen concentrations like in the pouch and do the dunnarts have any specialised fetal haemoglobins?

Overall, this is a very unique and well put-together study, describing a very interesting comparison in brain development between two distantly related mammals. These findings provide important insights into brain development and the order of events which lead to the patterning of the mammalian brain.

NCOMMS-23-25454 "Non-uniform temporal scaling of developmental processes in the mammalian cortex." Paolino et al., Response to reviewers' comments

We are grateful to all three reviewers for providing considered and helpful comments on our study, which we have now incorporated in a new version. We think that our manuscript has substantially improved in terms of clarity and impact of the findings. Point-by-point responses to specific questions and comments are detailed below:

Reviewer #1 (Remarks to the Author):

In this paper, Paolini A. and colleagues generate an extensive comparative analysis between FTD (a marsupial) and mouse (an eutherian) cortical development. This work is based on a large and detailed developmental timeline alignment between the two species, allowing developmental stage-based comparison. By combining neuronal birthdating techniques and in utero electroporations, the authors detail the pace of neuronal production, migration and maturation in both species. The study shows that the same developmental processes tend to occur faster in marsupials, suggesting a neoteny in eutherian cortical development and setting the ground for the complexification of neocortical structures. The authors relate this more rapid pace to the absence of intermediate progenitors, a key cell type to produce late-born neurons in eutherian species. All findings are nicely summarized in a final figure.

This study tackles an important evolutionary question and is of broad relevance. Points of discussion and suggestions are presented below:

Reviewer 1 comment 1 (R1.1): In the text, the terms basal progenitors and intermediate basal progenitors are used in an interchangeable manner. The terminology should be streamlined. More importantly, the lack of IPs in the FTD seems to be at the root of all the other observations, yet this point is not well presented in the text, results and discussion. Because of its mechanistic nature, I strongly recommend that the authors emphasize this point more, including in the title.

We have now amended the text throughout the manuscript to streamline the terminology to refer to “basal intermediate progenitor cells” in all cases except when referring to cell types/studies that could include both basal radial glia and basal intermediate progenitor cells, in which case we use the term “basal progenitor cells”. We have also now expanded our discussion of the role of basal intermediate progenitor cells (lines 753-772) which also

include newly refined conclusions (prompted by R1.8) that basal intermediate progenitor cells are likely not the only mechanism underlying these timing differences. Given this adjustment to our conclusions, to maintain the focus of our findings, and for the sake of brevity/character restrictions specified by the journal, we have chosen not to add this to the title.

R1.2: Figure 1: It is striking how thin L2/3 appears in panel J. In the somatosensory barrel cortex of the mouse, L2/3 and L4 have roughly the same width. With a DAPI staining it is difficult to appreciate layer limits. In order to quantify or at least display this, it would be better to use layer-specific markers.

The original brain sections that we used for these quantifications were co-labeled with the layer-specific markers SATB2 and CTIP2. We have repositioned our markers of layer width based on these markers' expression and have obtained new data for panels l, m, and n, incorporating this information in the new version. During this process, we realised that we had made a typographical error in previous panels m and n, where L2/3 and 4 were shown as separate, and L5/6 were combined, where it should have been the other way around based on our quantifications (i.e. L2/3/4 combined and L5 and L6 separate). Our original conclusions, however, were correctly based on the layer labeling that was in our original data – that the upper layers (L2/3/4) are significantly larger in dunnarts and the deeper layers (L5 and 6) are significantly larger in mice. Therefore, these conclusions have not changed. Our new data incorporating SATB2 and CTIP2 labeling also confirm this pattern, and we have added new qualitative images to panels j and k to help illustrate the species-specific layering differences. The figure legend (Fig. 1) and relevant methods sections have also been updated accordingly.

R1.3: Figure 3: In order to display differences across developmental time points and species, for panel C, it would be necessary to normalize cortical width to 1. From Figure 1i, it is clear that marsupial cortex is thinner in adults. This would also help to better illustrate the evolution of cell-type proportions across development.

We chose to represent our violin plots as absolute width measurements because the normalised distributions/widths of compartments can be inferred from the absolute width, but the absolute size information would be lost if only percentage width of the cortex was shown. However, we agree with the reviewer that the additional representation of the data normalised by cortical width helps to illustrate some aspects of developmental progression that are more difficult to appreciate from the representation of absolute widths, and have therefore now

added this representation in a new panel (e) of Supplementary Figure 3. We have also provided this normalized representation for the data in the other violin plot for comparative consistency, which is now Supplementary Figure 3p.

R1.4: Figure 5: please comply with a homogeneous terminology for intermediate progenitors.

All terminology is now consistent by using the term “basal intermediate progenitor cells” as per the remark R1.1 above.

R1.5: Figure 5: as for figure 3, please normalize cortical thickness to 1 in order to compare cell-type proportions and distribution across laminae. Although unlikely, the observed bimodal distribution of mouse PH3+ cells could also be observed in marsupials but the representation doesn't allow this to be assessed. A comparable scale could help to reinforce the author's findings.

As noted in our response to R1.3, we agree that this additional representation in a comparable scale is a useful addition to the manuscript, and have added another set of violin plots with the same data as in Fig. 5g, the difference being that the results are represented as a percentage of total cortical width (see new Supplementary Fig. 5a). We believe that this additional representation reinforces our findings that the distribution of PH3+ cells is divergent between species.

R1.6: Figure 5: single cell RNA seq dataset, are IPs present? Or are Tbr2+ cells exclusively non-cycling?

We have not presented a single cell RNAseq dataset in this article, and none has so far been published for a developmental marsupial cortex. Here we have shown histologically that TBR2+ cells in the subventricular zone of dunnarts are not undergoing mitosis (i.e. they do not express the mitotic marker PH3, Fig. 5j-l), and predominantly do not express cell-cycle markers (PCNA negative, Fig. 5m-o). While future scRNAseq studies may help to gain further insight into the specific identity of these cells, as well as providing candidates for the mechanistic changes underlying these interspecies differences, we do not think that this approach is necessary to support the conclusions of the current manuscript.

R1.7: Figure 6: It is interesting that between the two species, there is an opposing dynamics in contralateral targeting: medial to lateral in mouse, vs lateral to medial in

FTD. It appears, however, difficult to compare the projection dynamics of mouse medial callosal neurons to FTD medial intracortically projecting neurons. Is this intracortically projecting population present in mice? In FTD, this medial projection may be independent of anterior commissure projections. Perhaps reinforcing the discussion on the emergence of medial callosal projecting neurons in mice would be enough.

In mice, a similar population of intracortically projecting neurons has been reported; these were described as frontal projections, which are formed during postnatal development (between P2 and P8) compared to callosal projections that form during embryogenesis (Mitchell and Macklis, 2005). Otherwise known as association projections, these axons have been shown to course medially through the grey matter from the primary somatosensory cortex to the primary motor cortex, with the majority originating from dual-projection neurons that connect to both the contralateral and ipsilateral cortices (Oka et al., 2021). The distinct targets of these populations are precisely why we find their comparison interesting, especially given the finding that the order of “lateral axons first, medial axons second” is conserved between species despite these differences (Fig. 6 a-f). We have now detailed this in the results section to add additional information about the mouse intracortical medial projections, as well as making the rationale for this comparison more explicit by speculating that the medial intracortically projecting population in the dunnart may have provided a scaffold that was hijacked by callosal neurons during eutherian evolution (lines 461-467).

R1.8: Figure 7: the absence of intermediate progenitors can indeed explain the extension of the pace of migration and maturation of indirectly-born neurons in the mouse cortex. However, directly born neurons should exhibit similar developmental tempos. This could be assessed by comparing the fates of electroporated neurons that remain unlabeled following chronic BrdU injections (i.e. born from cells that never underwent a second round of division after electroporation) to those that are labeled. Flashtag could be used instead of electroporation for this purpose too.

We thank the reviewer for this insightful suggestion that the comparison of direct versus indirect neurogenesis populations within mouse could provide further insight into the interspecies differences we describe here. To address this, we have added a new analysis of our original data (new Supplementary Fig. 2c and f). This analysis is based on the premise that over 10% of neurons are born via direct neurogenesis in the mouse cortex (Huilgol et al., 2023) and they will likely constitute the most advanced cells compared to the neurons born

via indirect neurogenesis. Therefore, we averaged the position coordinates of the most advanced 2% of cells in the cortex of both mice and dunnarts (to ensure that this subsample would be well within the subpopulation of approximately 10% of mouse neurons born via direct neurogenesis) and compared this across collection stages. As was indicated by the original leading-edge analyses (Fig. 2b and d) this comparison revealed that the most advanced 2% of dunnart cortical cells are significantly more advanced than those of mouse (Fig. Supplementary Fig. 2c and f). We have updated the text to convey this result (lines 186-187), added a new discussion paragraph explicitly drawing the conclusion that a lack of basal intermediate progenitor cells is likely not the only mechanism contributing to the difference in timing (lines 753-772), and refined our conclusion in the abstract (lines 24-25). We believe that this addition has helped to clarify the mechanistic hypotheses generated by our findings.

R1.9: Figure 8: relating to the waiting period of DL contralateral projections, does this promote homotopic projections specificity? In mice there is some degree of areal refinement across postnatal development. Is this also the case of the FTD or it is prevented by the presence of this waiting period?

We have previously comprehensively studied the adult patterns of interhemispheric connectivity in marsupials, reporting shared features of contralateral connectivity across Theria, including broadly homotopic targeting as well as specific connectivity hubs (Suarez et al., 2018). Given this, and the evidence of waiting periods for all axons entering the cortex in rats and cats (referenced in lines 627-632), we hypothesise that it is likely that the phenomenon of deeper layer versus upper layer waiting periods is shared by eutherians and marsupials but has been too experimentally difficult to evidence in eutherians thus far, due to the need for multiple in utero electroporations at distinct ages targeting equivalent areas in individual embryos. We have now added reference to the adult shared features and made our hypothesis more explicit (lines 632-636).

Reviewer #2 (Remarks to the Author):

The authors are to be congratulated for a much-needed and long-awaited empirical contribution to an understanding of the scaling of developmental processes underlying the generation of the cortex in mammals. The methods are well-chosen, optimal for their purposes, and the quantification is fine.

R2.1: As a general editorial comment, I'd note that their results sections and discussion

suffer from the alphabet-soup of acronyms generic to present developmental neurobiology. While they do a good job in their summary titles of each section describing what will be covered, I would suggest injecting more redundancy throughout (for example, the function of a particular tracer, names and significance of layers or developmental compartments) to help the non- or not-yet expert along. Similarly, the figures, constrained by the need to pack 40 into 4, seem more to attend to packing than communication. For example, the graphs which compare timing on a maturational scale for mouse versus dunnart would naturally stack vertically for best comparison; it might even save some space. Now, they appear to have no particular system. Reduction in graphs/figure and more parallel construction would help.

We thank the reviewer for this point. Indeed, we were constrained by the word and figure limits. We have inserted more redundancy of term usage, including spelling out acronyms, throughout the manuscript, but perhaps more could be done if space is allowed. Regarding the orientation of figures, we have consistently used the convention of plotting time along the x axis as the primary system of presentation and have optimised presentation of all meaningful data in as little space as we can. We will work with the editor to address these points further.

R2.2: The only substantial criticism I have is with the word “non-uniform”, which in the title is applied to temporal scaling of developmental processes. “Scaling” implies a comparison, or some sort of normalization, but just what temporal scaling might be is neither commonly done nor understood very well, in my experience. What about developmental processes could be scaled and what “non-uniform” means in this context might be better spelled out.

We thank the reviewer for raising this point, which gives us an opportunity to explain our main findings and their relevance more clearly. One possible scenario is that for species with dramatically different developmental timescales (e.g. the mouse and the 3 times slower-developing dunnart) distinct processes of cortical development will scale equivalently with the overall whole-body developmental time course, as quantified by a developmental staging system (i.e. all three processes will be 3 times more protracted in dunnart than mice in absolute terms). This equivalence is what we refer to as “uniform”. Thus, our main finding is that while one of the processes that we study (cell birth/lamination) does scale in this way (i.e. it is similarly timed between species according to an equivalent staging system and therefore it is “normalised” by the overall developmental stage which works as an anchoring

timeline for all developmental events), the other processes on which we focus (migration and axon extension) do not scale in this way. This difference is what we refer to as “non-uniform temporal scaling”, i.e. these processes do not all uniformly scale with reference to the equivalent staging system and, therefore, to each other. To clarify this, we have now explicitly set out the aforementioned alternative hypothesis in the introduction (lines 93-99), as well as making some of the wording more aligned with the title in the abstract (line 23).

R2.3: For example, a great deal is known about the intrinsic variability of neurogenesis, with respect to its absolute time, – absolute time to generate a neuron is variable across species (and thus sometimes temperature), brain region and cell class; the duration of a cycle of neurogenesis typically extends with maturational stage, but not all cell cycle components are lengthened uniformly, etc. The authors do a good job in dissecting several of the possible components, but are hampered in laying out an explanation of "non-uniform temporal scaling", by the fact that the product of neurogenesis, the cortex, is chosen to be as similar as possible for this investigation. To produce a similar end number of neurons, but over a duration three times as long, what possibilities for alteration of cell cycle kinetics are even possible? Considering absolute time, not only layer-specified targeting of neuron number, what has scaled to produce such uniformity? It would be interesting to compare (not new studies, just extant literature) controls for absolute time, not absolute neuron output – comparisons to the eutherian ferret or cat cortex (or retina). Relative and absolute extensions of time might be informative, as in the Cebus versus Aotus retina (Dyer et al., PNAS '09).

We chose mice and dunnarts precisely because they have comparable cortices, but very distinct developmental timescales, in order to investigate how distinct processes can scale over different ontogenetic tempos to produce comparable outcomes. We have now expanded our explanation of the known differences in cell cycle length that likely underlie these different absolute developmental timescales between marsupials and eutherians in the introduction, as well as citing extant literature comparing the timescales of brain development in the cat and short-tailed opossum (lines 47-51). We have avoided comparisons with other tissues (e.g. retina) for the sake of brevity and to avoid potential confusion due to lineage- (e.g. eutherian-specific constraints) and/or tissue-specific (e.g. retina versus neocortex) potential confounds. Moreover, we have now included additional descriptions of the possible scenarios relating to what might be conserved versus the potential for variation (for example

axon growth versus cell cycle dynamics, respectively) to result in similar brain outcomes (lines 772-781).

R2.4: When it comes to neuronal migration and axon extension, comparatively very much less is known about absolute rates of progress, by species or maturational stage. Both developmental phenomena would seem to have fewer “targets” for modification (i.e. base stem cell number, the duration of the phases of the cell cycle, and alteration of the number of cells cycling, and more!). The authors’ results suggest that in fact, neither cell migration nor axon extension reflect species-of-origin (could the rate be constant?), and thus stack up “early” in the maturational state of the slower maturing dunnart. An interesting suggestion is that these “mismatches” are buffered systemically -- by an SVZ as an outflow region for cases where neurogenesis exceeds migration capacity (Charvet & Striedter 2011: Causes and consequences of expanded subventricular zones, Eur. J. Neurosci), or by a “waiting period” for tardy migration as shown here, and in citations. The authors might be presenting a central case of “phenotypic integration” or “evolvability”, the present paper a first step in building such a systemic understanding. The unlikelihood of vast accelerations in the pace of such axon outgrowth, given the “rodent” (both types) to monkey range should be presented to make clear what “non-uniform” would mean!

We thank the reviewer for highlighting these aspects of our study. We have now added an extra paragraph to the discussion (lines 772-781) outlining the possible mechanisms that may underlie our results, referencing some of the literature alluded to by the reviewer, including the Charvet & Striedter 2011 article and additional references comparing migration and axon extension rates between eutherian species to strengthen these points. We agree that the phenomena we are describing likely represents a key example of evolvability, and have now explained this further in the discussion (lines 818-820).

R2.5: Overall, while I appreciate they are trying to present a consistent definition of a maturational-stage based temporal scaling, failing to provide a concrete anchor of just what might be scalable in a developmental mechanism would help comprehension. Providing it would give the paper more impact.

The “anchor point” we are using is our published whole-body developmental staging system, which is based on highly conserved vertebrate features such as development of the eyes. We have now added further discussion about what may and may not be scalable (lines 753-781).

The three specific processes of cortical development that we report do not align uniformly to this staging system: one does (cell birth/specification) while the other two are relatively faster in the dunnart than the mouse.

Reviewer #3 (Remarks to the Author):

The paper by Fenlon et al provides very unique insights into Understanding the rules and limitations of timing in neocortical development and how this is mediated by changes in gene expression. The unique use of a marsupial model and exquisite tools developed in this species to visualise these dynamics is commendable and shed new insights into the development and evolution of the mammalian brain.

This is an extremely careful anatomical study – rarely seen with such precision. Novel insights into heterochrony in brain development between mice and dunnarts. The paper flows on from a careful staging series of the dunnart head which describes and aligns developmental milestones in neurogenesis with the mouse model. This underpins the work here, showing the differential emergence and migratory paths of marsupial neurons relative to the developmental stage.

The study is well described, the data are beautifully and clearly presented. And the outcomes are clear.

R3.1: One of the only points I had was surrounding the rationale for these differences. In the discussion the authors make the statement ‘cortical neurogenesis occurs entirely postnatally in dunnarts versus entirely prenatally in mice’... ‘Despite this, it is likely that dunnart joeys receive more sensory stimulation than mice at equivalent stage in utero’ Presumably olfactory and gravitational forces are important for navigation to the pouch ie. before birth, do these early events drive changes in early brain progenitors? Once in the pouch, external stimulation is somewhat muted and the only stimuli they need to respond to is suckling? I wonder if the authors can comment on / expand the uniqueness of this requirement, the marsupial brain at birth and this might underpin some of the findings here?

This is an excellent suggestion that we did not raise in the original discussion. We have now included speculation that advanced neurodevelopment not only might be required for early behaviours such as entry into and occupancy of the pouch, but that this early sensorimotor activity might also be driving advanced neurodevelopment (lines 797-800).

R3.2: The oxygen scenario is also an interesting one. Can the authors state if they are suggesting the dunnart has a reduced or increased oxygen environment relative to a mouse in utero? What are oxygen concentrations like in the pouch and do the dunnarts have any specialised fetal haemoglobins?

We do not know whether the dunnart has reduced or increased levels of brain oxygenation relative to the mouse *in utero*. Based on the published patterns of mouse oxygen manipulation during gestation, where lower oxygen levels produce fewer mitoses of basal progenitor cells, it is possible that dunnart brains are less oxygenated, but there are many contributing factors that complicate this, some of which the reviewer mentions. We have expanded the discussion to elaborate on what is known about these physiological differences and have specifically hypothesised that dunnarts have lower oxygen levels than mice (lines 800-816).

Overall, this is a very unique and well put-together study, describing a very interesting comparison in brain development between two distantly related mammals. These findings provide important insights into brain development and the order of events which lead to the patterning of the mammalian brain.

REVIEWERS' COMMENTS

Reviewer #1 (Remarks to the Author):

I am satisfied with the answers of the authors and their updates to the manuscript. This is a really nice piece of work illustrating the importance of developmental timings in setting up brain architecture.

Reviewer #2 (Remarks to the Author):

The authors have responded well to the concerns raised in the first review, within the confines of the necessary page and figure requirements of the journal.

Reviewer #3 (Remarks to the Author):

The authors have addressed all my questions in manuscript.